# Social attraction in *Drosophila* is regulated by the mushroom body and serotonergic system

Yuanjie Sun [1,2,6], Rong Qiu[1,4,6], Xiaonan Li[1,2], Yaxin Cheng [1,2,5], Shan Gao [1,2], Fanchen Kong [1,2], Li Liu[1,2] & Yan Zhu [1,2,3 ✉]

Sociality is among the most important motivators of human behaviour. However, the neural mechanisms determining levels of sociality are largely unknown, primarily due to a lack of suitable animal models. Here, we report the presence of a surprising degree of general sociality in *Drosophila*. A newly-developed paradigm to study social approach behaviour in flies reveal that social cues perceive through both vision and olfaction converged in a central brain region, the γ lobe of the mushroom body, which exhibit activation in response to social experience. The activity of these γ neurons control the motivational drive for social interaction. At the molecular level, the serotonergic system is critical for social affinity. These results demonstrate that *Drosophila* are highly sociable, providing a suitable model system for elucidating the mechanisms underlying the motivation for sociality.

---

[1] State Key Laboratory of Brain and Cognitive Science, Institute of Biophysics, Chinese Academy of Sciences, 15 Datun Road, Beijing 100101, China. [2] University of Chinese Academy of Sciences, Beijing 100049, China. [3] Advanced Innovation Center for Human Brain Protection, Capital Medical University, Beijing, China. [4] Present address: New England Biolabs (Beijing) LTD., No.1 Wang Zhuang Road, Beijing, China. [5] Present address: Shenzhen Science Museum, Shangbu Road, 1003 Shenzhen, China. [6] These authors contributed equally: Yuanjie Sun, Rong Qiu. ✉email: zhuyan@ibp.ac.cn

Social affiliation is a fundamental behaviour in many species. In social animals, individuals in a population tend to associate in groups, and, in the case of humans, form cooperative societies. Moreover, social motivation deficits are common phenomena in mental disorders, including autism, depression, and anxiety disorder[1–4]. Understanding sociality and related disorders require a clear understanding of the evolutionary roots and neural substrate of the desire for social affiliation. Such an understanding would also help to elucidate how complex features, such as emotions and desires, emerge from networks of neurons in general.

Similarly to humans, many non-human animals exhibit a natural tendency to approach and investigate unfamiliar conspecifics, as reported in fish[5], mouse[6], and rat[7]. Social approach behaviour was measured initially by observing mice interacting freely in an open field[8] and later in a three-chambered assay[9]. Deficits in the social approach exhibited in mice models of mental disorders, such as autism[10], depression[11], schizophrenia[12], and anxiety disease[13]. However, the mechanisms determining levels of sociality are largely unknown, and systematically revealing the neural substrates of social approach behaviour would require a high-throughput approach to effectively screen a large number of genes or different strains.

As a basic instinct, sociality is reported to exist in insects, with well-known examples of highly social insects, such as honeybees, ants, and wasps. In contrast, *Drosophila* are typically thought to be largely solitary, exhibiting very limited social behaviour. Male-initiated types of communication have been extensively studied in fruit flies, including those for mating purposes, such as aggression[14,15], courtship[16,17], and copulation[18,19]. Nevertheless, transient social interactions between fruit flies within a group exhibit dynamic patterns[20], and flies aggregate to form organized networks[21,22]. Experienced flies have been reported to transfer knowledge about oviposition sites and parasitoid threats to naive flies[23], suggesting that *Drosophila* are capable of certain types of sociality beyond the purposes of mating[24]. However, the prevalence, intensity, and motivation of social interactions, as well as the underlying neural circuitry, remain ill-defined. The current study provides evidence that neurons in central brain regions of *Drosophila* regulate the tendency for social affiliation, a prerequisite for advanced social interactions. These findings provide a promising basis for elucidating the neural root of sociality.

## Results

**Strong social attraction in *Drosophila*.** To quantify the tendency for social interaction in *Drosophila*, we developed a high-throughput social approach paradigm. Inside a shallow circular chamber, 10 flies were tethered to one half of the chamber to serve as attractor flies (attractors). Subsequently, free-walking flies (hereafter referred to as subject flies) were released into the chamber, and their two-dimensional distribution over time was video-recorded and analysed with programs developed in-house (Fig. 1a, Supplementary Movie 1). Regardless of the type of social interaction, any physical interaction requires two animals to be close to each other; therefore, the amount of time a free-moving fly spends near immobilized attractor flies reflects the motivation for social interaction. As shown in Fig. 1b, in an empty chamber, female wild-type flies, *Canton-S* (*CS*), exhibited no bias to either side of the chamber. However, when attractor flies were tethered in one half of the chamber, free-walking *Canton-S* flies spent significantly more time on the same side as attractors, regardless of the sex of the attractors (Fig. 1b), suggesting that flies have a strong tendency to associate with other flies.

Because fruit flies prefer to walk along the chamber wall, a behaviour known as thigmotaxis[25–27] (Supplementary Fig. 1A–D),

we chose to analyse the distribution of subject flies inside a smaller area (a circle 5.5 mm away from the chamber wall) when quantifying sociability (Supplementary Fig. 1A). A preference index (PI) was devised to describe flies' tendency for sociality (see the Materials and Methods section for details, also movie S1). For example, on average *Canton-S* females appeared on the side containing male attractors ~75% of the time, yielding a PI score of 0.5 (PI = 0.75–0.25, Fig. 1c). Further characterization revealed that the PI value did not change significantly over 4 h (Supplementary Fig. 1E), and was not affected by the number of free-moving flies (Supplementary Fig. 2A, B), suggesting a stable social attraction. Analysis of single flies revealed that they displayed a tendency to "repeatedly" explore (Supplementary Fig. 2C) or stay near the tethered flies (Supplementary Fig. 2D) for various durations. We also found that free-moving flies were attracted to model flies constructed from Fimo polymer clay, which were the same size as *Canton S* female flies and had wings from real flies (Supplementary Fig. 1F), and food bars. Notably, the desirability toward to normal flies was substantially higher than that to Fimo flies and a food bar (Supplementary Fig. 1G). These results indicated that social motivation is a strongly purposive behaviour in flies.

Importantly, when agar bars with dimensions similar to those of flies were glued to a chamber in place of attractor flies, they failed to attract the free-moving flies, suggesting that the attractive action of subject flies is goal-oriented, instead of purely exploratory, and is performed to move close to other flies, but not non-fly objects (Supplementary Fig. 1G). Furthermore, dead attractor flies elicited similar PI values as living flies, indicating that feedback and interaction signals originating from attractor flies are not necessary for social approach in *Drosophila* (Supplementary Fig. 1H).

**Robust and ubiquitous social attraction.** Next, we set out to identify potential factors influencing social attractions, such as sex, physiological status, and genetic background. When testing the level of attractiveness of flies of each sex, we found that both male and female flies were readily attracted by immobilized flies, regardless of their sex, with female attractors exerting a slightly higher level of attraction on subject males (Fig. 1c). This similarity in the strength of social attraction suggested that our behavioural paradigm identified a relatively general motivation for sociality, instead of specific motivation for reproduction typically initiated by males. As sex did not influence social approach in our paradigm, we used female flies for our experiments.

Physiological status, such as satiety or mating state, can affect social interactions or function as a driving force for certain social behaviours. However, we found that neither age nor mating status affected the strength of the social approach (Supplementary Fig. 3A, B). Starvation slightly promoted social motivation (Supplementary Fig. 3C). Taken together, these data suggested that social approach behaviour was robust against physiological status.

In addition to *Canton-S*, several common lab strains (*Oregon-R*, *Berlin-k*, *w1118*, and *yw*) exhibited a similar strength of social approach to *Canton-S* flies (Fig. 1d), suggesting that social attraction was not simply influenced by genetic composition.

We further evaluated whether social attraction is a general trait among *Drosophila* species. As shown in Fig. 1e, the wild-type flies of five *Drosophila* species were strongly attracted by their conspecifics, suggesting that social affiliation is ubiquitous in *Drosophilidae*. We also found that *Canton-S* flies were attracted by immobilized wild-type flies of five *Drosophila* species, respectively (Supplementary Fig. 4A), Furthermore, all species exhibited approach behaviour toward *D. melanogaster*, but to

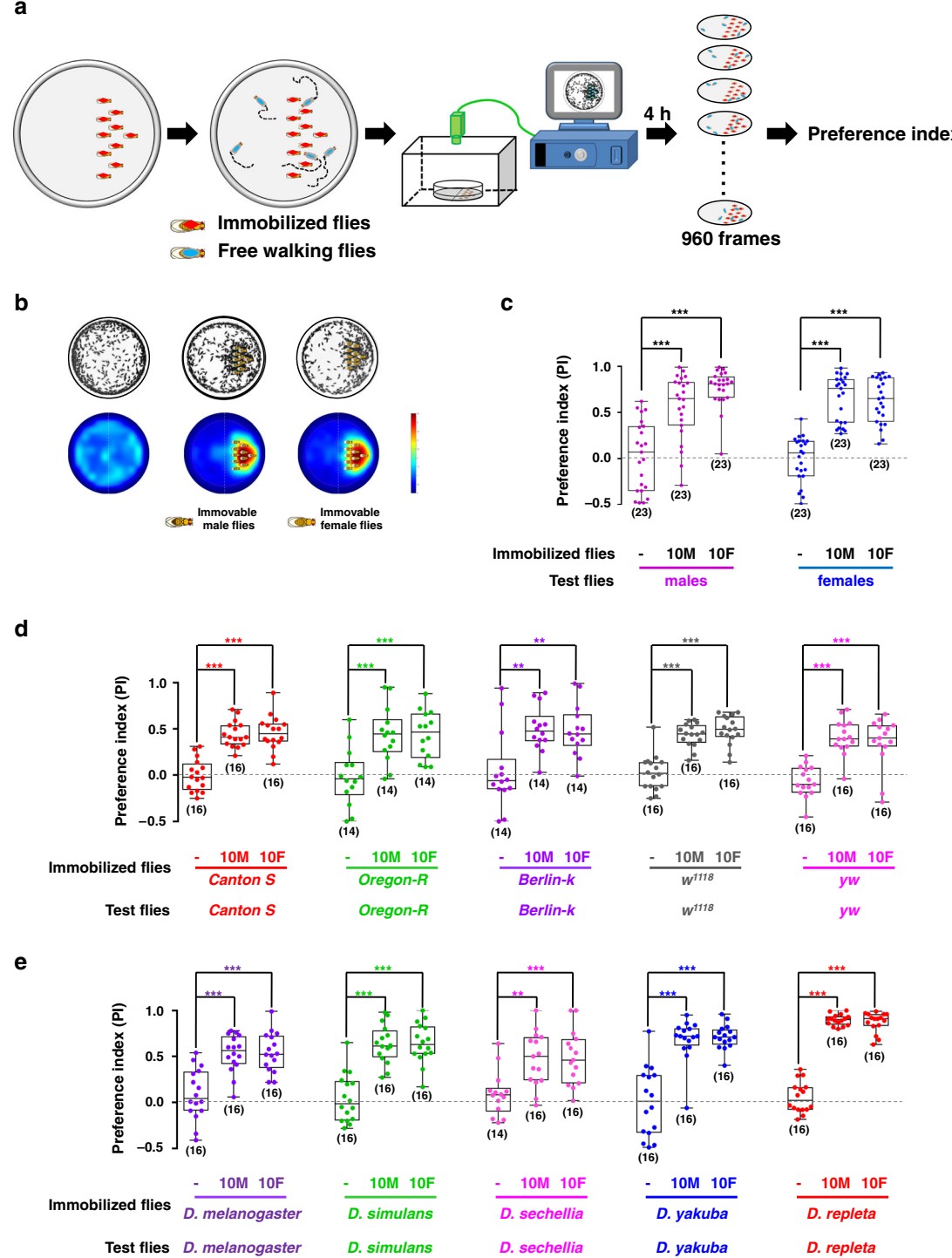

**Fig. 1 Social attraction in *Drosophila melanogaster*. a** Procedure for quantifying social approach behaviour. Immobilized attractor flies were tethered to the right side of a shallow dish while the left side was empty. The distribution of free-walking flies in the dish was tracked using a video camera, and the preference index was calculated (see also Video 1). **b** Free-walking flies were strongly attracted to immobilized flies. Top panels: the overlays of 960 frames from a 4-h test period. Bottom panels: accumulated distributions of free-walking flies. **c** *Canton S* flies of both sexes were similarly attracted by male and female attractor flies. **d** Genetic background had minor effects on social attraction. The strains are *Canton-S*, *Oregon-R*, *Berlin-K*, *w*[1118], and *yw*. **e** Wild flies of different species of *Drosophilidae* exhibit social attraction. The species tested were *D. melanogaster*, *D. simulans*, *D. sechellia*, *D. yakuba*, and *D. repleta*. The results are presented as a box and whisker plot; the whiskers indicate the minimum and maximum, the box includes the 25th–75th percentile, and the line in the box indicates the median of the data set. Statistical analysis: one-way ANOVA followed by Dunnett's test was used for comparison of all groups vs. control group, and t-test for comparisons between only two groups (**e**). **$P < 0.01$, ***$P < 0.001$.

different degrees that correlated well with the evolutionary distance (Supplementary Fig. 4B). Thus, social attraction in *Drosophila* is robust and evolutionarily conserved.

**Past experiences affect social attraction**. Socially isolated flies were previously reported to be more aggressive, with abnormal circadian activities and a short lifespan[28,29]. We found that, compared with group-reared flies, flies raised in isolation exhibited a weaker tendency to approach attractors (Supplementary Fig. 5A). The remaining tendency of social affiliation in the singly raised flies indicated that social attraction is largely an innate response in *Drosophila*.

To further examine the plasticity of social approach behaviour, we conditioned wild-type flies with attractors surrounded by denatonium, a potent repulsive chemical for fruit flies. In the test after training for 6 h, wild-type flies exhibited a decrease in social approach toward standard attractors without denatonium (Supplementary Fig. 5B), demonstrating that, despite the strength and robustness of social approach behaviour, it also exhibits a high degree of flexibility.

Next, we labelled and mixed male wild-type flies with a mutant strain, *fruitless*, which exhibits intense male–male courtship behaviour[30]. Interestingly, being pursued erroneously for 2 days by mutant males for courtship, wild-type flies exhibited dramatically decreased PI scores toward wild-type attractor flies in subsequent tests, indicating a lower motivation for sociality (Supplementary Fig. 5C). Taken together, these data indicated that motivation for social interaction is reduced by negative past social experiences.

**Social attraction uses multiple sensory cues**. To identify the major sensory signals mediating social attraction, we eliminated various sensory modalities, including olfactory, visual, auditory, and gustatory sense in subject flies. For this set of experiments, wild-type flies were chosen as attractor flies (the cue provider), while the free-moving flies (cue perceiver) were deprived of specific sensory modalities by genetic mutations and physical methods (see the "Methods" section for details). The contribution of olfactory inputs was tested in *Orco* mutant flies, which lack odorant perception[31], as well as in wild-type flies in which the primary olfactory organs, third antennal segments, and maxillary palp, were surgically removed[32] (Fig. 2a). The role of vision in social attraction was determined in wild-type flies (tested in darkness) and vision-impaired mutant flies, *norpA*[P33] (tested under light) (Fig. 2b). Auditory sensory function was eliminated by either *iav*[1] mutation[33] or by surgically removing the arista, the primary auditory organs (Fig. 2c). Finally, we used flies with a mutation eliminating all gustatory receptors[34], *Poxn*[Δm22], to investigate the contribution of gustatory sensory function (Fig. 2d). Surprisingly, under all conditions the attractiveness of wild-type flies to "sensory deprived" flies persisted, suggesting that flies use redundant sensory inputs across multiple modalities for social affiliation.

To test our hypothesis, we simultaneously removed either two or three different sensory inputs. Removing visual and olfactory sensory functions together significantly decreased the level of social approach (Fig. 2e and Supplementary Fig. 6), whereas the removal of other pairs of sensory modalities did not affect social approach (Supplementary Fig. 7). Furthermore, flies with three sensory modalities removed, such as visual-auditory-gustatory and olfactory-auditory-gustatory functions, maintained normal social approach behaviour (Supplementary Table 1). Overall, these data demonstrated that flies use both visual and olfactory cues to exhibit attraction toward other flies. The redundancy of sensory inputs is consistent with the robustness of social

approach behaviour, increasing the chance of a fly recognizing and socializing with conspecifics, even under challenging situations.

Insects use chemical information to find conspecifics for social interactions, such as aggregation, courtship, and aggression. In *Drosophila*, the male-specific pheromone *cis*-vaccenyl acetate (cVA) is considered to act as a long-range signal to mediate aggregation, and as a short-range signal to induce aggressive and sexual behaviours[35,36]. To test whether cVA is involved in social approach behaviour, we tethered virgin *Canton-S* flies as attractors. In the dark, the virgin females, which were devoid of cVA, exhibited decreased attraction to walking flies (Supplementary Fig. 8A). However, the attraction of virgin females in darkness was not abolished (Supplementary Fig. 8A), suggesting that cVA was one of the pheromones mediating social attraction in this situation. We further tested the receptor neurons of cVA for their roles in perception of social cues. There are two kinds of cVA receptor neurons in Drosophila: *Or67d* neurons (labelled by *Or67d-Gal4*) and *Or65a* neurons (labelled by *Or65a-Gal4*). It was previously reported that *Or67d* neurons serve as receptors for tracking male-deposited landmarks[37]. Flies with cVA receptor neurons silenced by TNT were tested in the dark for their social affiliation ability (Fig. S8B). The social approach response was strongly supressed when *Or67d* receptor neurons were silenced, while silencing the *Or65a* receptor neurons did not impact social attraction (Fig. S8B). This suggests cVA signalling mediated by the *Or67d* receptor neurons is important for perception of the conspecific social cues, including its essential roles in tracking the deposits of other flies.

**Social motivation requires central brain neurons**. We next investigated the corresponding brain regions underlying this robust and conserved motivation for sociality. To identify the higher brain centres responsible for motivating free-moving flies to approach attractors, we suppressed the activities of neurons in specific brain regions with tetanus toxin (TNT) using the GAL4/UAS system (Fig. 3a)[38]. We surveyed neuronal populations in all major brain regions: the mushroom body (MB, Supplementary Fig. 9a–i), ellipsoid body (EB, Supplementary Fig. 9–n), fan-shaped body (FB, Supplementary Fig. 9o–s), suboesophageal zone (SEZ, Supplementary Fig. 9t-v), protocerebral bridge (PB, Supplementary Fig. 9n), and antennal mechanosensory and motor centre (AMMC, Supplementary Fig. 9w–y). As shown in Fig. 3b, the level of social approach was dramatically reduced after silencing specific MB neurons in subject flies, whereas suppressing the activities of neurons in other brain regions failed to exert similar effects (Fig. 3c–g). Interestingly, not all MB > TNT combinations resulted in a decrease in sociality. This behaviour was only abolished following expression of TNT in the γ lobe neurons of the MB while silencing neurons in other MB lobes (α/β and α′/β′), exhibiting no effect on social attraction behaviour (Fig. 3b and Supplementary Table 2). Because the five drivers we used to label the Kenyon cells (KC) contributing to the gamma lobes also exhibited expression elsewhere inside or outside MB, we further tested a set of split-Gal4 drivers that were highly specific for different types of KC in MB[39] (Supplementary Fig. 10). Social approach behaviour was substantially reduced only when expressing TNT in KC$_\gamma$ neurons (labelled by MB607B, MB419B, MB009B, or MB131B), whereas silencing other KC neurons (KC$_{\alpha/\beta}$: MB008B, MB477B, MB185B, MB594B or MB371B; and KC$_{\alpha'/\beta'}$: MB005B and MB463B) did not affect social attraction (Fig. 3h). Taken together, these findings strongly supported the notion that KC$_\gamma$ neurons are necessary for the motivation to approach conspecifics. To confirm the silencing results, we transiently silenced MB neurons with tub-Gal80$^{ts}$, MB > TNT in a temperature-

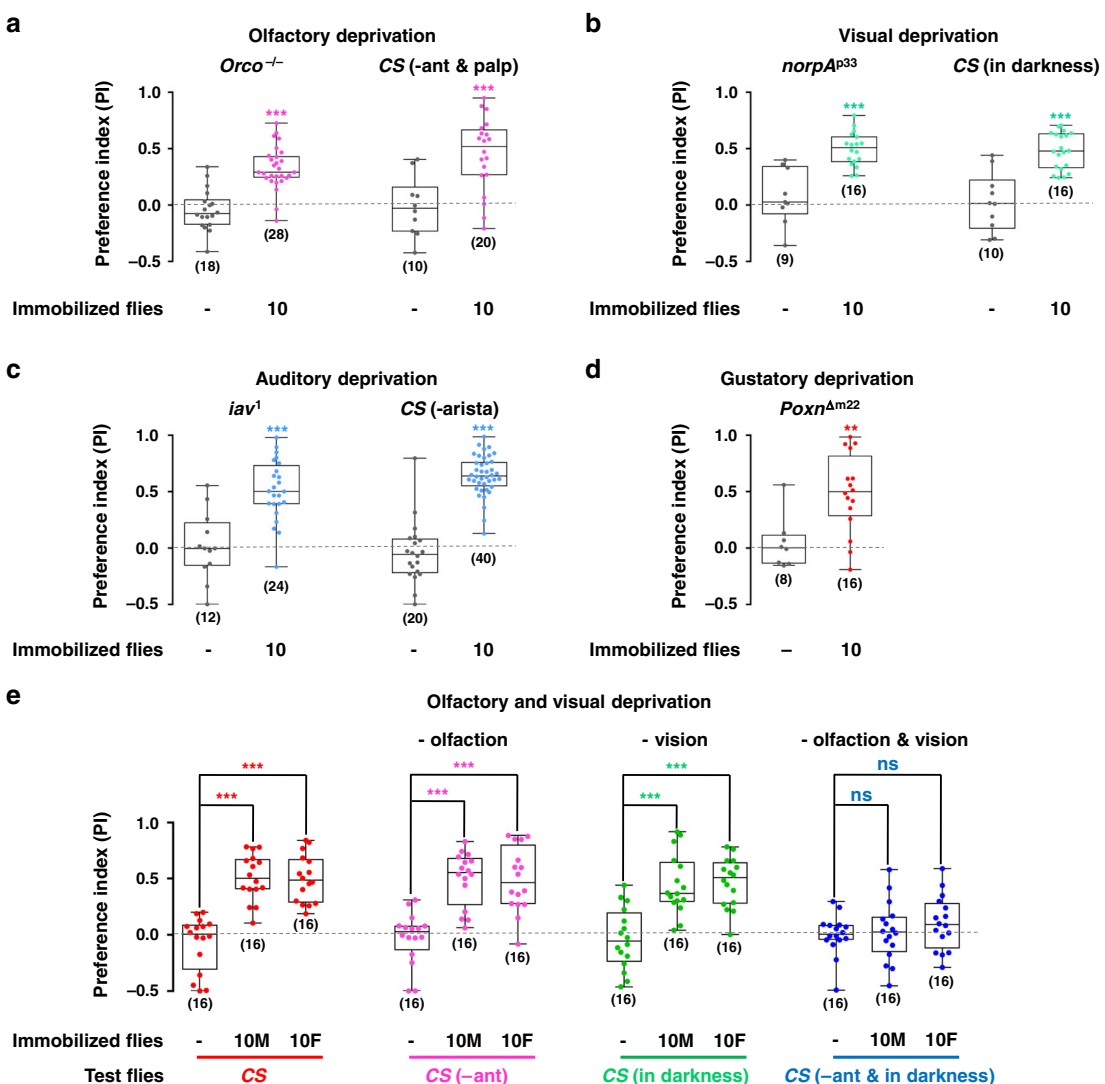

**Fig. 2 Olfaction and vision mediate social attraction in *Drosophila melanogaster*. a–d** Removing a single sensory modality was not sufficient to block social approach behaviour. **a** Deprivation of olfaction through using *Orco* mutants or *Canton S* lacking antennae (-ant) and maxillary palp (-palp). **b** Deprivation of vision through using *norpAp33* mutants or testing *Canton S* in the dark. **c** Deprivation of auditory sense through using *iav1* mutant or *Canton S* without arista. **d** Deprivation of gustatory sense through *PoxnΔm22* mutants. **e** Simultaneously removing the olfactory and visual sense, by testing *Canton S* lacking antennae (-ant) in the dark, eliminated social approach behaviour. $n = 16$. Results are presented as a box and whisker plot; whiskers indicate the minimum and maximum, the box includes the 25th–75th percentile, and the line in the box indicates the median of the data set. Statistical analysis: t-test for two group only comparisons (**a–d**) and one-way ANOVA followed by Dunnett's test for multiple comparisons (**e**). ns: $P > 0.05$, **$P < 0.01$, ***$P < 0.001$.

dependent manner (Fig. 3i), revealing that the level of social approach was dramatically decreased by adult-stage inactivation of $KC_\gamma$ neurons, but not $KC_{\alpha/\beta}$ or $KC_{\alpha'/\beta'}$ neurons (Fig. 3j).

Silencing $KC_\gamma$ neurons exhibited no significant impact on vision, olfaction, or locomotion, suggesting that the visual and olfactory inputs for social approach remained intact under such conditions (Supplementary Fig. 11A–C). Single fly tracking also revealed that flies with silenced $KC_\gamma$ neurons spent less time in the area containing attractor flies (Supplementary Fig. 12). Therefore, it appears that silencing $KC_\gamma$ neurons reduces the motivation for continuing social interactions.

If $KC_\gamma$ neurons participate in promoting social approach behaviour, their activities might be regulated by social encounters. We used a nuclear factor of activated T-cells (NFAT)-based neural tracing method, CaLexA, to visualize neurons that were activated upon encountering other flies[40]. To correlate specifically the social encounters with the activity of $KC_\gamma$ neurons, we identified a specific line, *R72B08-Gal4*. *R72B08-Gal4* drove

expression in multiple regions of the brain (Supplementary Fig. 13A, B). However, in the MB, it specifically labelled $KC_\gamma$ neurons (Supplementary Fig. 13C). Social approach behaviour was sharply reduced when silencing *R72B08-Gal4*-labelled neurons with TNT (Supplementary Fig. 13D). As expected, compared with socially isolated flies, group-reared flies exhibited higher fluorescence intensity in both the calyx region (Fig. 4a, b) and the lobe region (Supplementary Fig. 14A). We also compared fluorescent signals in other regions between group-reared flies and socially isolated flies. The signals in either the α/β lobe region (Supplementary Fig. 14B) or the calyx region (Supplementary Fig. 14C) were not statistically significant. Taken together, our results strongly suggested that $KC_\gamma$ neurons are a critical component of the neural circuits promoting social affiliation.

Next, we investigated whether activating the $KC_\gamma$ neurons would upregulate social affiliation in flies. As shown in Fig. 4c, flies with optogenetically activated $KC_\gamma$ neurons (labelled with either *NP1131-Gal4* or *R72B04-Gal4*) exhibited higher levels of

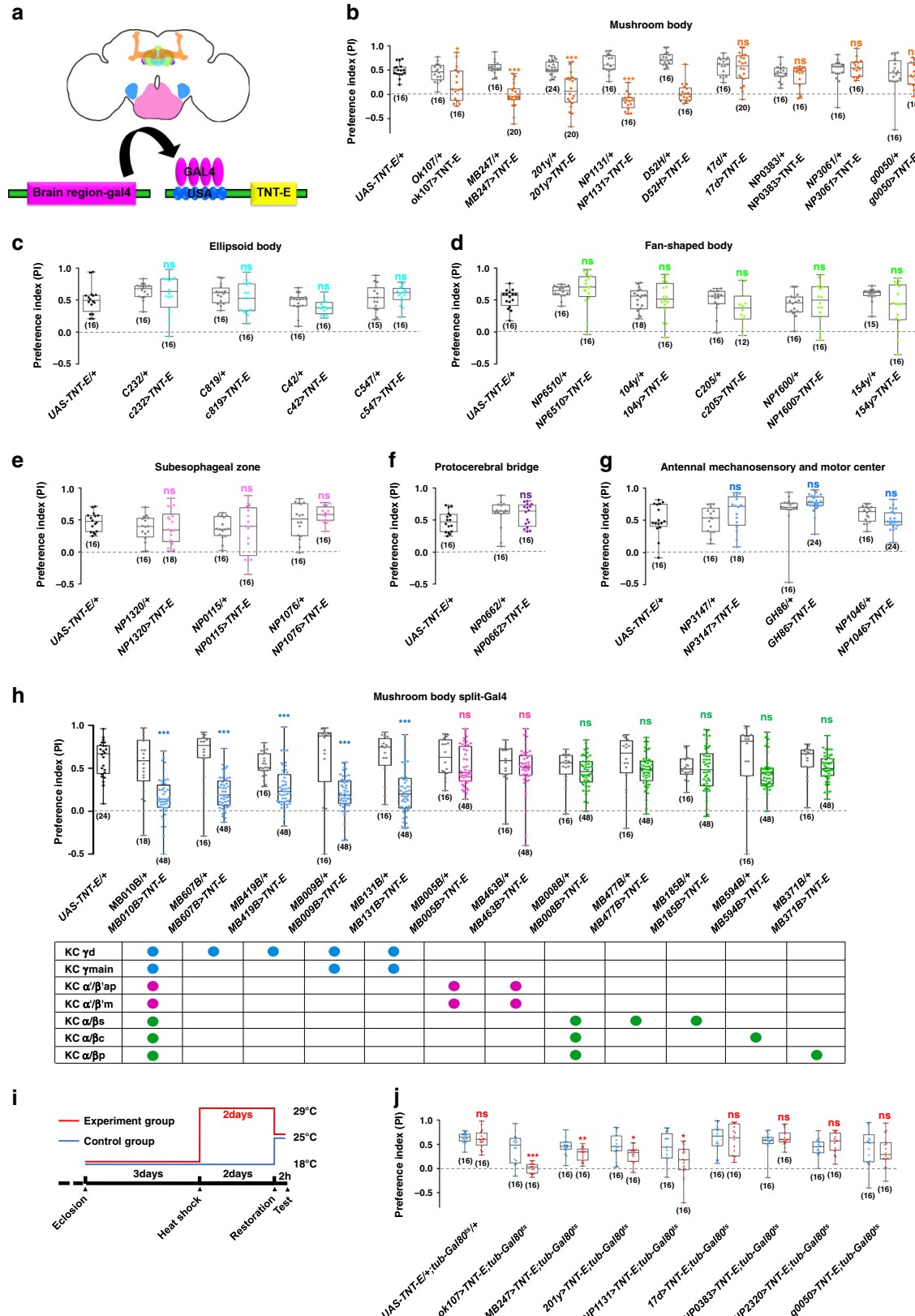

social approach to wild-type attractors than did the controls. When inter-species attraction was tested, *D. melanogaster* flies found *D. repleta* flies exhibited the least attractive among all species surveyed (Supplementary Fig. 4B). Interestingly, with their KC$_\gamma$ neurons activated, the *D. melanogaster* flies exhibited

higher tendency to affiliate with *D. repleta* than did the controls (Fig. 4d). The results demonstrated that the activity of KC$_\gamma$ neurons regulated the flies' motivation for social affiliation, suggesting that KC$_\gamma$ neurons function as a critical integration centre for sociality.

**Fig. 3 The γ lobe of the mushroom body mediates social motivation. a** A diagram to show different brain regions in the adult fly (top) and the inactivation scheme by the GAL4/UAS system (bottom). Different Gal4s drove the expression of TNT in specific neurons in selected brain regions, including the mushroom body (**b**), ellipsoid body (**c**), fan-shaped body (**d**), suboesophageal ganglia (**e**), protocerebral bridge (**f**), and AMMC region (**g**). The social approach level was dramatically altered by expressing TNT in neurons labelled in ok107, MB247, 201y, D52H, and NP1131 lines (**a**), but not other neurons (**a**–**g**). **h** The social approach levels were affected when silencing populations containing $KC_\gamma$ neurons in MB010B ($\alpha/\beta + \alpha'/\beta' + \gamma$), MB607B ($\gamma_d$), MB419B ($\gamma_d$), MB009B ($\gamma_{d+main}$), and MB131B ($\gamma_{d+main}$), but were normal when silencing the $KC_{\alpha/\beta}$ and $KC_{\alpha'/\beta'}$ neurons ($\alpha/\beta$: MB008B, MB477B, MB185B, MB594B, and MB371B; $\alpha'/\beta'$: MB005B and MB463B). Different KCs occupy distinct layers in the lobes as indicated (a: anerior; d: dorsal; p:posterior; m: medial; c: core; s: surface). Blue, pink, and green circles represents expression in γ, α'/β', and α/β lobe, respectively. **i** A scheme for inactivating neural activity in a selected time window by elevating the temperature to allow the expression of TNT. Flies containing UAS-TNT-E, tub-Gal80ts, and various Gal4s were kept at 18 °C for 3 days after eclosion. Subsequently, flies were kept at 29 °C for 2 days. The social approach tests were conducted after restoration at 25 °C for 2 h. **j** The results of selective inactivation via controlling temperature. The social approach levels were decreased in ok107, MB247, 201y, and NP1131. Results are presented as a box and whisker plot; the whiskers indicate the minimum and maximum, the box includes the 25th–75th percentile, and the line in the box indicates the median of the data set. Statistical analysis: unpaired *t*-test. ns: $P > 0.05$, *$P < 0.05$, **$P < 0.01$, ***$P < 0.001$.

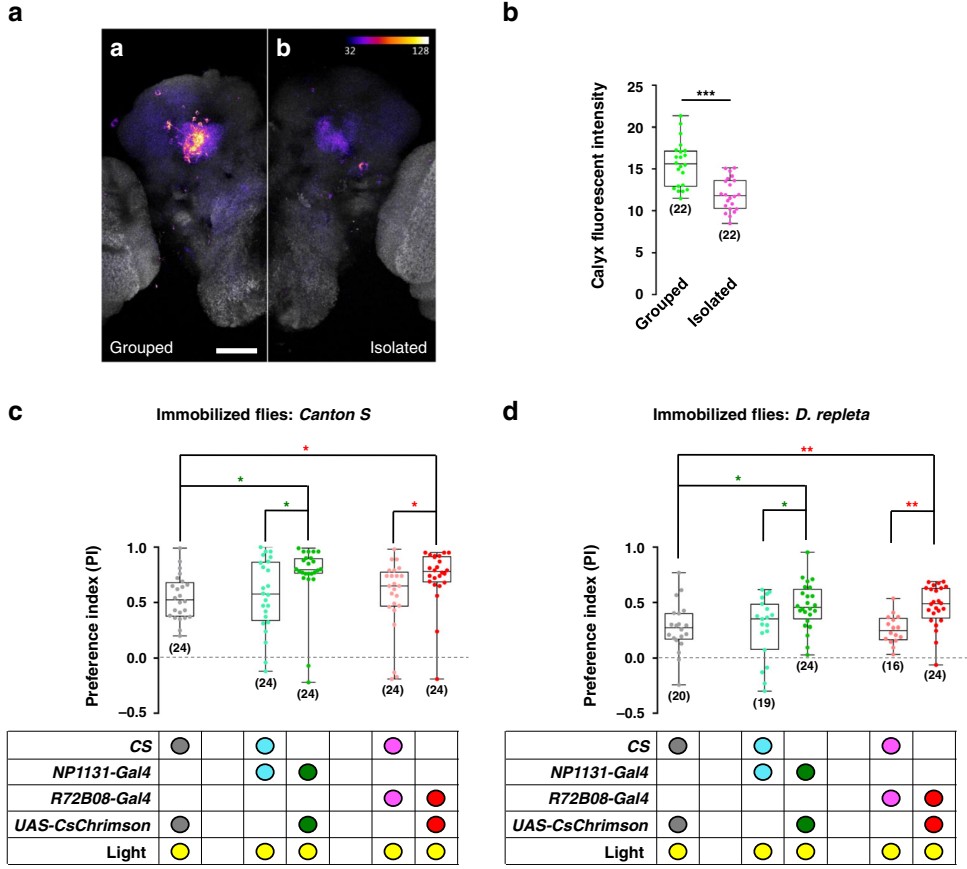

**Fig. 4 Activation of $KC_\gamma$ neurons increases social motivation in *Drosophila*. a** Social experience increased neural activity in $KC_\gamma$ neurons. Comparison of confocal images of the $KC_\gamma$ neurons in flies bearing *R72B08-Gal4, UAS-mLexA-VP16-NFAT, LexAop-CD2-GFP*, and *LexAop-CD8-GFP-2A-CD8-GFP* transgenes revealed that flies reared in a group (Grouped, **a**-a) exhibited more activity than flies reared individually (Isolated, **a**-b). Scale bars = 50 μm. **b** Quantification of signal intensity in the calyx regions of grouped and isolated flies. **c** Optogenetic activation of *NP1131-Gal4* or *R72B04-Gal4* labelled KC neurons elevated levels of social attraction between the flies of the same species. Attractor flies were wild-type *Canton S*. **d** Activating the *NP1131-Gal4* or *R72B04-Gal4* labelled $KC_\gamma$ neurons also increased attraction between species. Attractor flies were wild *D. repleta*. The results are presented as a box and whisker plot; the whiskers indicate the minimum and maximum, the box includes the 25th–75th percentile, and the line in the box indicates the median of the data set. Statistical analysis: one-way ANOVA followed by Dunnett's test was used for comparison of all boxes vs. control box. *$P < 0.05$, **$P < 0.01$, ***$P < 0.001$.

**Dissecting circuitry of social attraction.** We next investigated the neural circuit connecting sensory inputs with the central brain KC neurons. Because of the functional redundancy between vision and olfaction in social approach, we sought to identify critical neurons in the visual pathways in flies deprived of the sense of smell by surgically removing antennae (Fig. 5a). Under these "anosmic" conditions, silencing the F5 neurons (in layer 5 of the FB) abolished social approach behaviour (Fig. 5b),

suggesting that F5 neurons are involved in relaying visual information to the MB.

To identify essential neurons in olfactory pathways for social attraction, we tested animals kept in darkness during the experiment, to exclude any visual cues (Fig. 5c). Under these conditions, the levels of social attraction were significantly decreased following the silencing of the α/β surface neurons (Fig. 5d, e d, e), whereas blocking the other class of $KC_{\alpha/\beta}$

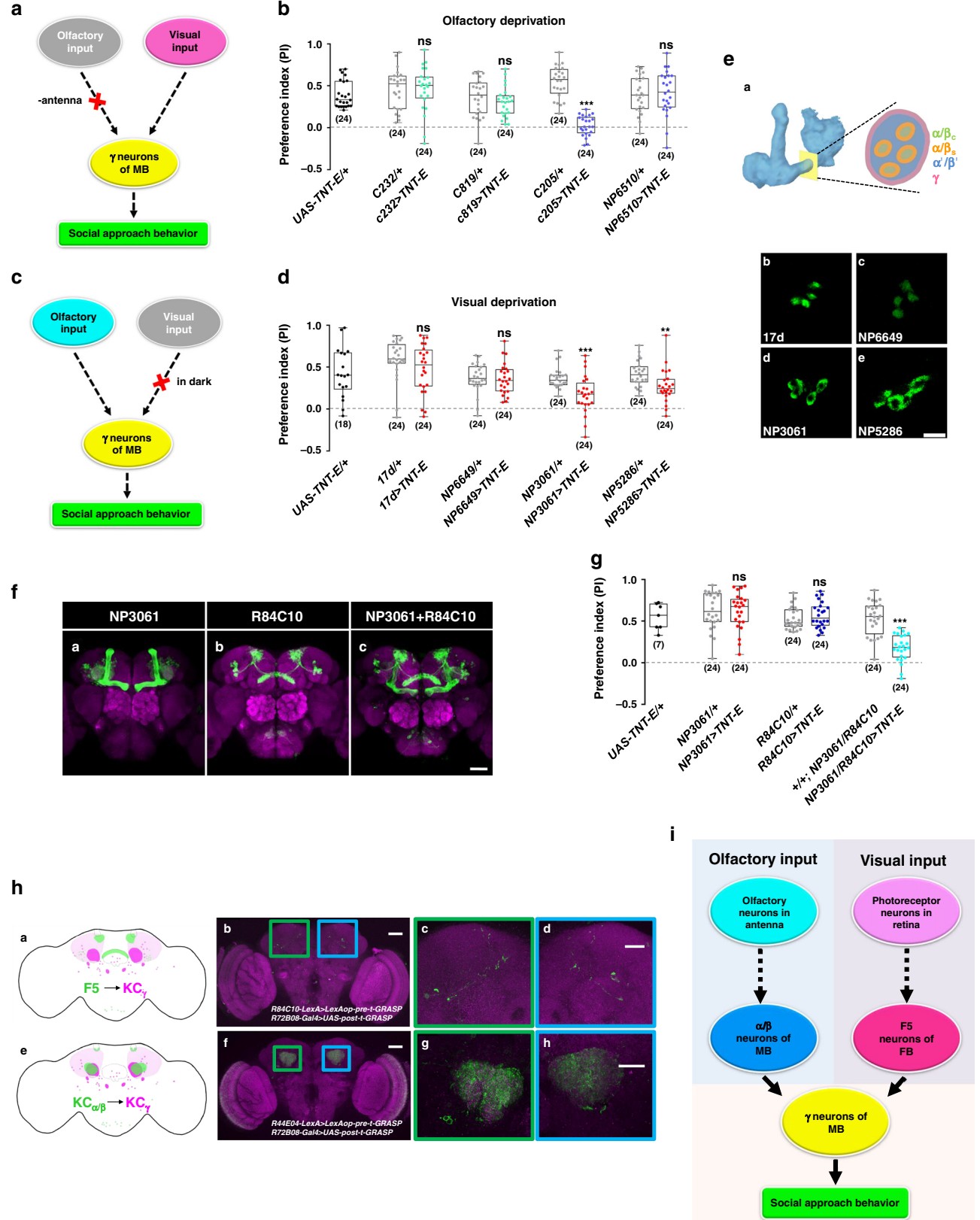

neurons, namely α/β core neurons, had no visible effects (Fig. 5d, e b, c), suggesting that KC$_{α/β}$ surface neurons are responsible for delivering olfactory information.

When both F5 neurons and KC$_{α/β}$ surface neurons were simultaneously silenced, flies with intact antennae exhibited greatly reduced social attraction under regular illumination

(Fig. 5f, g, Supplementary Fig. 15A, B). Furthermore, simultaneously blocking KC$_{α/β}$ surface neurons and other neurons in the FB did not result in impairment of social approach behaviour (Supplementary Fig. 15C, D).

To facilitate the reconstruction of the neural network of social attraction, we investigated the synaptic structures of KC$_γ$, F5, and

**Fig. 5 Mapping the neural circuit of social approach behaviour.** Schematic (**a**) and results (**b**) of investigating the visual pathway in social interaction, via inactivating visual neurons in the flies without olfactory organs. Schematic (**c**) and results (**d**) of investigating the olfactory pathway in social interaction, by testing flies in the dark with blocked $KC_{\alpha/\beta}$ neurons. **e** Two classes of $\alpha/\beta$ neurons exhibit different cross-section appearances at the peduncle. (**e**-a) Schematic of a cross-section of the peduncle of MB. Cross-sections of the expression patterns of 17d (**e**-b), NP6649 (E-c), NP3061 (**e**-d), and NP5286 (**e**-e) at peduncle, visualized by GFP. **f** NP3061 (**f**-a) and R84C10 (**f**-b) show specific expressions in the mushroom body and fan-shaped body, respectively. **g** Simultaneous inactivation of $\alpha/\beta$ surface neurons (NP3061) and F5 neurons (R84C10) reduced social approach behaviour, while inactivation of either population alone did not affect social interaction. **h** Visualization of the synaptic connections between F5 neurons, $\alpha/\beta_s$ neurons, and $\gamma$ neurons in the calyx by t-GRASP. (**h**-a) Schematic of the locations of the presynaptic structure labelled by R84C10 (green) and postsynaptic structure labelled by R72B08 (magenta). (**h**-b) t-GRASP signals indicated contacts between $KC_\gamma$ neurons (*R72B08-Gal4*) and F5 neurons (*R84C10-LexA*) in the calyx region (coloured box). (**h**-c and d) Magnified views in the boxed regions of (**h**-b). (**h**-e) Schematic of the locations of the presynaptic structure labelled by R44E04 (Green) and postsynaptic structure labelled by R72B08 (magenta). (**h**-f) t-GRASP signals showed the contacting sites between $KC_{\alpha/\beta s}$ neurons (*R44E04-LexA*) and $KC_\gamma$ neurons (*R72B08-Gal4*) in the calyx region (coloured box). (**h**-g and h) Magnified views in the boxed regions of (**h**-f). The neuropil was counterstained with the antibody against nc82 (magenta). Scale bars = 50 μm, except in (**e**: 10 μm) and (**h**-c, d, g, and h: 25 μm). **i** A diagram to show the neural circuit mediating social attraction. Results are presented as a box and whisker plot; the whiskers indicate the minimum and maximum, the box includes the 25th–75th percentile, and the line in the box indicates the median of the data set. Statistical analysis: unpaired *t*-test. ns: $P > 0.05$, **$P < 0.01$, ***$P < 0.001$.

$KC_{\alpha/\beta}$ surface neurons. We found that the postsynaptic terminals of $KC_\gamma$ neurons and the presynaptic structures of F5 neurons were enriched mainly in the calyx region (Supplementary Fig. 16A -D), strongly suggesting that F5 and $KC_\gamma$ neurons can form direct synaptic connections. To test this possibility, we visualized the connections between the F5 neurons (labelled by *R84C10-LexA*) and $KC_\gamma$ neurons (labelled by *R72B08-Gal4*) using the targeted green fluorescent protein (GFP) reconstitution across synaptic partners (t-GRASP) method, which enhanced its specificity for synaptic contact sites via a targeting strategy to label the presynaptic terminals with GFP11 and the postsynaptic regions with GFP1-10[41]. Only when the F5 and $KC_\gamma$ neurons separately expressed the complementary halves of GFP, stable labelling was observed near the calyx region (Fig. 5h b–d), while the other combinations of various transgenic components served as negative controls (Supplementary Fig. 17A). In another set of t-GRASP experiments, the F5 neurons were labelled by *R84C10-Gal4* and $KC_\gamma$ neurons were labelled by *92F10-LexA* (Supplementary Fig. 17B). Swapping the Gal4 and LexA drivers produced similar distribution patterns of synaptic sites (Supplementary Fig. 17B-f to -h). The synaptic connection from F5 neurons to the $KC_\gamma$ neurons confirmed the behavioural results of silencing these neurons.

To investigate signalling from $KC_{\alpha/\beta}$ surface neurons to $KC_\gamma$ neurons, we first surveyed the potential downstream synaptic targets of $KC_{\alpha/\beta}$ surface neurons using *trans*-Tango, a method of anterograde transsynaptic tracing[42]. In flies bearing the *NP3061-Gal4* driver (labelling $KC_{\alpha/\beta}$ surface neurons) and the *trans*-Tango components, $KC_{\alpha/\beta}$ surface neurons were visualized to innervate to $\alpha/\beta$ surface lobes as well as $\gamma$ lobes (Supplementary Fig. 18A). Later connectivity patterns indicated that the $KC_{\alpha/\beta}$ surface neurons transmit information to $KC_\gamma$ neurons, which we further tested with the t-GRASP method. In the negative controls, the combinations of different transgenic components did not generate detectable signals (Supplementary Fig. 18B). Only when the $KC_{\alpha/\beta}$ surface neurons (labelled by *R44E04-LexA*) and $KC_\gamma$ neurons (labelled by *R72B08-Gal4*) separately expressed the complementary halves of GFP, intense labelling was observed near the calyx region (Fig. 5h f–h), indicating that $KC_{\alpha/\beta}$ surface neurons form synaptic connections with $KC_\gamma$ neurons. Similarly, another set of t-GRASP experiments on $KC_{\alpha/\beta}$ surface neurons (labelled by *NP3061-Gal4*) and $KC_\gamma$ neurons (labelled by *R92F10-LexA*) produced similar distribution patterns of synaptic sites (Supplementary Fig. 18C).

The calyx is subdivided into the main and accessory calyces, which receive different inputs. It has previously been shown that olfactory inputs project to the main calyx while visual stimuli project to the accessory calyx[43,44]. In accordance with these

previous studies, our t-GRASP experiments here indicate that the social cues of distinct sensory modalities are represented by different KC subsets in the subdomains of the calyx. Synaptic sites between the $KC_\gamma$ neurons and F5 neurons were detected in the accessory calyx (Fig. 5h b–d and Fig. S17B f–h), while the synaptic sites between the $KC_\gamma$ neurons and $KC_{\alpha/\beta}$ surface neurons were detected in the main calyx (Fig. 5h f–h and Fig. S18C g–i). Overall, the data suggested the presence of a microcircuit in the central brain that promotes social attraction (Fig. 5i).

**Social motivation depends on serotonin.** Previous studies indicated that serotonergic neurons affect social interaction in human[45] and rodents[11]. Furthermore, infection by *P. locustae* in locusts suppresses the generation of aggregation pheromones. This, in turn, reduces the production of the neurotransmitter serotonin that initiates gregarious behaviour[46]. In *Drosophila*, the serotonergic system is involved in both aggression[47] and courtship[48]. Thus, we investigated whether the serotonergic system has a positive effect on sociality in general. As shown in Fig. 6a, TNT-induced silencing of serotonergic neurons (labelled by *TPH-Gal4*) resulted in decreased social approach behaviour. As dorsal paired medial (DPM) neurons have been suggested as a source of serotonin to the peduncles and lobes[49] (Supplementary Fig. 19A and B), we tested the role of DPM neurons in social motivation. Social approach behaviour was dramatically decreased when silencing DPM neurons with TNT (Supplementary Fig. 19C). Next, we restricted the number of TNT-inactivated neurons using different Gal80 strains (Fig. 6a). The expression patterns shown in Fig. 6b–d suggested that the remaining serotonergic neurons in central brain regions were responsible for normal social approach behaviour.

We next investigated which of the five serotonin receptors (5-HT1A, 5-HT1B, 5-HT2A, 5-HT2B, and 5-HT7) are involved in social affiliation. Interestingly, although serotonergic neurons are distributed broadly throughout the brain, the expression of *5-HT1B* is mainly limited to within MB and EB in braion (Fig. 6e). Using a pan-neural driver (*elav-Gal4*) to express RNAi in the brain, we found that reducing the expression levels of the 5-HT1B receptor, but not of the other 5HT receptors, yielded flies that exhibited low motivation for social approach (Fig. 6f).

We then investigated whether 5-HT1B receptors are required in $KC_\gamma$ neurons for social drive. *5-HT1B-Gal4* mainly drove expression in $KC_\gamma$ neurons of the MB and neurons in the EB (Fig. 6e, h a–c). When we introduced MB-specific Gal80 (*MB-Gal80*) into the flies, specifically eliminating Gal4-driven expression in the MBs (Fig. 6h d–f). Although expressing the 5-HT1B RNAi broadly (by *elav-Gal4*) or specifically (by *5-HT1B-Gal4*)

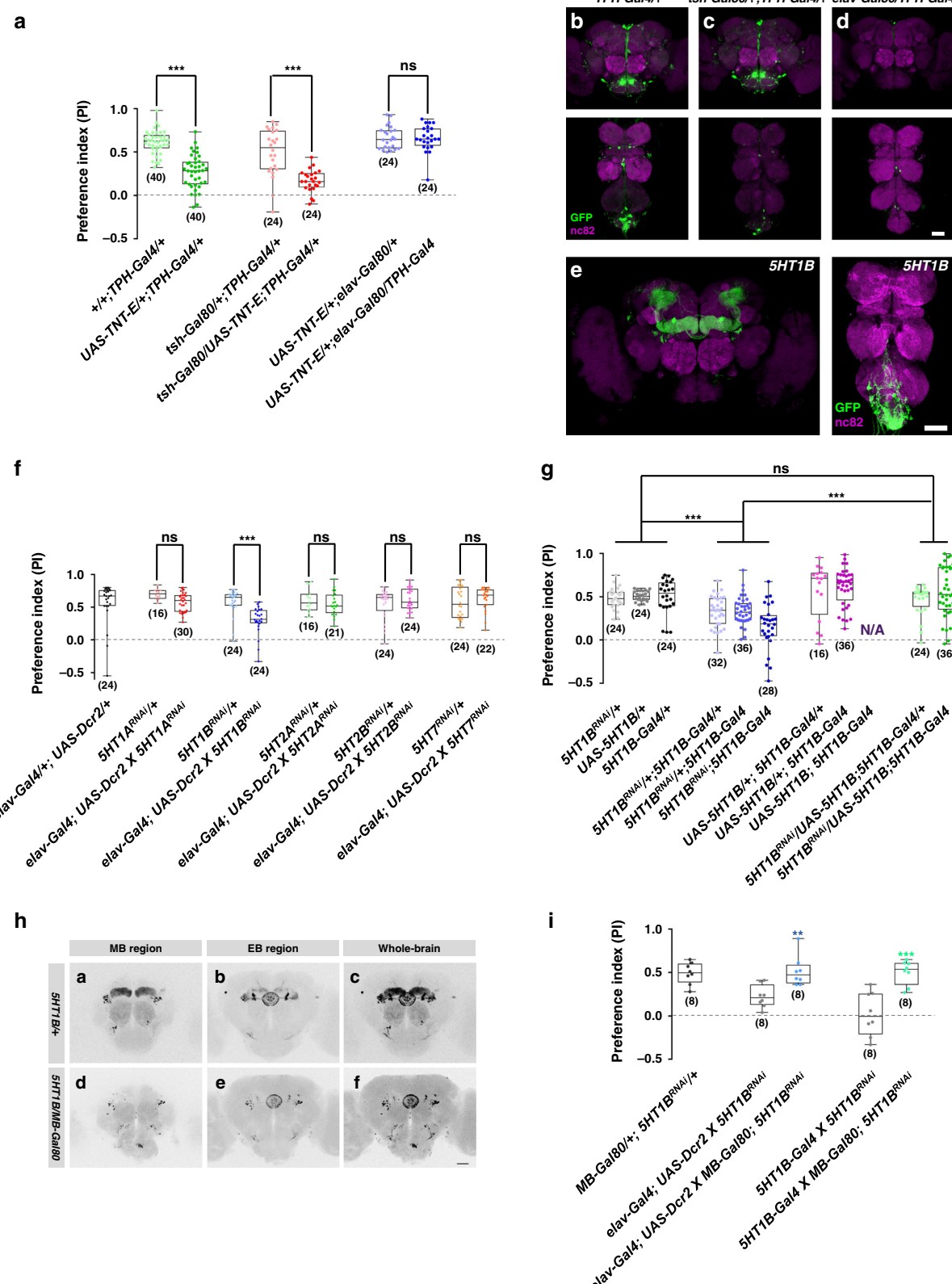

decreased social approach behaviour (Fig. 6f, g), we found that only preventing 5-HT1B RNAi expression in KCγ neurons restored social approach behaviour (Fig. 6i), indicating that normal social approach requires 5-HT1B receptors in the MB. Moreover, restoring the expression level of 5-HT1B completely rescued the deficit phenotype (Fig. 6g).

Taken together, our data strongly suggested that serotonin plays a key role in promoting sociality, namely through the 5-HT1B receptor in KCγ neurons. Similarities at the behavioural and molecular levels between Drosophila and mammals raise the question of whether the motivation for social affiliation is a well-conserved feature.

**Fig. 6 The serotonin system is required for social approach. a** Inactivation of all serotoninergic neurons by expression TNT in *TPH-Gal4* labelled neurons (green) resulted in decreased levels of social approach behaviour. This phenotype was rescued by the pan-neural expression of *Gal80* (*elav-Gal80*, blue), but not by the Gal80 expressed specifically in VNC (*tsh-Gal80*, red). **b** Expression patterns of *TPH-Gal4* in the brain and VNC. **c, d** Expression patterns of *TPH-Gal4* in the brain and VNC when combined with *tsh-Gal80* (**c**) and *elav-Gal80* (**d**). **e** Expression patterns of *5-HT1B-Gal4* in the brain and VNC. **f** Identifying serotonin receptors for social motivation. Knockdown with *5-HT1B* RNAi (blue) resulted in reduced social approach performance, but not with RNAi of other receptors: *5HT1A* (red), *5HT2A* (green), *5HT2B* (pink) and *5HT7* (yellow). **g** Restoring the expression of *5-HT1B* rescued the social approach defects from the *5-HT1B* knockdown. **h** Upper: the expression patterns of the *5-HT1B* receptor by *5-HT1B-Gal4* in different brain regions: mushroom body (MB) region (**h**-a), ellipsoid body (EB) region (**h**-b), and whole-brain (**h**-c). Lower: the restricted expression patterns of *5-HT1B-Gal4* in the presence of *MB-Gal80* in different brain regions: MB (**h**-d), EB (**h**-e), and the whole brain (**h**-f). **i** Comparison of the social approach behaviours in *5-HT1B* knockdown flies with (blue or green) and without (grey) the presence of *MB-Gal80*, which blocked the expression of *5-HT1B RNAi* in MB. Results are presented as a box and whisker plot; the whiskers indicate the minimum and maximum, the box includes 25th–75th percentile, and the line in the box indicates the median of the data set. Scale bar = 50 μm. Statistical analysis: unpaired *t*-test (**a, f, i**) and one-way ANOVA followed by Dunnett's test (**g**). ns: $P > 0.05$, \*\*$P < 0.01$, \*\*\*$P < 0.001$.

## Discussion

The present findings suggest that social attraction is ubiquitous and robust in *Drosophilids*. The results revealed that the γ lobe of the MB functions as an integration centre for social cues. $KC_\gamma$ neurons are influenced by social experience and promote social attraction. At the molecular level, the current results provide strong evidence that serotonin is critical for social affinity in *Drosophila*[45].

Sociality is one of the most intriguing and complex behavioural traits of animals, ranging from insects to humans. Sociality in social ants and bees has been characterized as eusociality[50]. Compared with social insects, fruit flies lack advanced forms of sociality, operating at the pre-social level[51]. The tendency of individuals to associate in groups is a crucial feature of sociality and a prerequisite for the development of advanced social interactions. Importantly, the group-forming tendency is the driving force behind sociality, and manifests through measurable behaviours.

Our paradigm focused on quantifying the internal drive of flies to actively seek social affiliation, instead of analysing the subsequent elaborated (if any) social interactions. The assay helped us to focus on the neural circuitry underlying this type of desire. The persistence of such behaviour over 4 h suggested that the observed behaviour was more than social investigation or social attraction. The social cues were both visual and olfactory, similar to the mechanism found in ants and honeybees[52,53]. Removing the ability to sense cVA decreased but did not abolish the behavioural response, indicating redundancy in olfaction for social cues. Notably, this general sociality is distinct from courtship and aggressive behaviour. In the current study, the results demonstrated that social attraction was not sex-specific, whereas courtship and aggression are typically initiated by males and involve a pair instead of a group. Furthermore, these behaviours involve different types of actions and use different neural circuits[14–19].

Groups of free-walking flies typically aggregate[21,54], and, through a highly dynamic process, may self-assemble into a cluster with regular spacing[22]. The current investigation helped to elucidate the attractive forces (olfaction, vision, and physiological and neural states) that brings individual flies together. Although the precise aetiological function of sociality in *Drosophila* is not immediately apparent, active social gathering may increase the likelihood of finding food and mating, as well as the spreading of information[24].

We identified the MB as the integration centre for social approach behaviour. The MB is a prominent structure in the central brain of insects and is thought to play a similar functional role as the hippocampus in mammals[55]. Social cues are transmitted to the MB through F5 neurons (visual pathway) and $KC_{\alpha/\beta}$ neurons (olfactory pathway), both of which function as upstream neurons of $KC_\gamma$ neurons. Furthermore, activity of $KC_\gamma$ neurons, which receive visual input[43], is required for social attraction. The basal activity of $KC_\gamma$ neurons is modulated by social experience, and forcibly manipulating their activity changes the motivation for social affiliation. The MB has been extensively studied for its role in learning and memory[56,57], decision-making[58], and, more recently, in circadian rhythm and sleep[59,60]. Interestingly, recent studies have shown that $KC_\gamma$ neurons are involved in reward-related learning and memory[61]. The current results suggest a new function for MB in the expression of sociality.

Sociality in mammals has been studied extensively, particularly in humans. However, the neural circuits underlying the fundamental social instinct remain elusive. Importantly, previous studies of memory formation, including the case of patient HM[62], demonstrated that an intact hippocampus is required for social memory in both mice and humans. Taken together with these earlier findings, the current results suggest that a potentially similar mechanism for sociality exists across the animal kingdom. Moreover, serotonin is known to be involved in regulating sociability in mammals[45]. In accordance with earlier studies, our results demonstrated that, in *Drosophila*, serotonin and its receptors are also required for sociability.

The interesting parallels of sociality at the behavioural, anatomical, and molecular levels between fruit flies and mammals suggest that *Drosophila* provides a promising animal model for studying the neural basis of sociality, as well as potentially facilitating our understanding of sociability impairments in humans at a level that is currently too challenging with other animal models.

## Methods

**Fly stocks**. Fly stocks were maintained at 25 °C and 60% humidity under a 12:12 h light/dark cycle. *Canton-S* flies[63] were used as a wild-type control in our experiments. Details of fly stocks are listed in the key resources table (Supplementary Table 3).

**Social approach analysis**. *Drosophila* were collected within 1 day of birth and maintained in regular food vials. We used 4- to 5-day-old flies for all behavioural experiments except for social isolation experiments (see below). The behavioural experiment was performed using a transparent 60-mm Petri dish filled with 1% agar to 5 mm below the lid. Ten same-sex flies acting as attractor flies were fixed at predefined positions on top of the agar in one half of the dish using glue. Five female flies acting as free-walking flies were then transferred into the dish. After 30-min of habituation, the movement of the flies in the dish was recorded for 4 h using a modified webcam at 15 s per frame. Social approach behaviour was defined as the proportion of times the free-walking flies entered the side containing the immobilized flies. For each frame, the performance index (PI) was calculated as follows: PI = (number of subject flies appearing on the side with tethered flies − number of subject flies appearing on the side without tethered flies) / total number of subject flies. The overall PI was the averaged PI value for the designated period. Social entrance index = (number of subject flies entering the side with tethered flies − number of subject flies entering the side without tethered flies) / total number of subject flies. Residence time = duration time of flies staying on the side with tethered flies.

**Denatonium conditioning**. The behavioural experiment was performed using two separated 35-mm Petri dishes (a training dish or a plain dish vs an empty dish, see below) in a transparent container (a 90-mm Petri dish with its bottom covered with 1% agar). A 35-mm Petri dish filled with 1% agar was designated as an empty dish. A plain dish was an empty dish with ten flies tethered on the agar surface. A training dish was a 35-mm Petri dishes filled with 6 mM denatonium diluted in 1% agar and had 10 flies tethered on the agar surface, serving for conditional training with denatonium. Before training, five free-moving flies were transferred to a container with one plain dish and one empty dish for 4 h to quantify their pre-ference before training. Subsequently, the five flies were transferred to a second container with one training dish and one empty dish for 8 h to obtain their pre-ference during the training phase. After training, the flies were immediately transferred to a third container with one plain dish and one empty dish for 4 h to evaluate their preference. The distributions the free-moving flies on the two 35-mm dishes were used to calculate the preference at each time point: PI = (number of free-moving flies on the dish with tethered flies − number of free-moving flies on the empty dish) / total number of free-moving flies on both dishes. The overall PI for each phase of conditioning was the averaged PI of the corresponding period.

**Male–male courtship conditioning**. $fru^M$ null males ($fru^{LexA}/fru^{4-40}$)[30] were collected at eclosion, housed individually for 4 days, then put together in groups of seven $fru^M$ null males with one $w^{1118}$ male for 4 days. The social approach behaviour of $w^{1118}$ males was assayed, as described in the Social approach analysis section above.

**3D image registration**. Fly brains were dissected and stained, and fluorescent images were collected with a confocal microscope in dual channel mode. One channel recorded the nc82 signal as counter-staining, while the second channel recorded the GFP signal. Each channel generated an XYZ image stack of the whole fly brain. The imaging data were then processed in Fiji[64] using 3D Viewer plugin[65]. The registration procedure was similar to previously described methods with minor modification[65]. First, both the stack of the female template brain[66] and the stack of nc82 channel were loaded into the 3D Viewer. Eight homonymous landmarks (two in each side of the PB, two at the conjunctions between the optic lobes and the central complex, two in each side of the suboesophageal ganglion and two at the tips of the alpha lobes of the MBs) were manually positioned on each of the two stacks. The rigid transformation matrix was then generated and applied to the stack of the GFP channel. Similarly, a second brain, in which GFP labelled a different brain region, was also registered onto the same template brain. Finally, the regis-tered GFP stacks and template brain were merged together to demonstrate a 3D spatial relationship between brain regions.

**Quantification of calyx fluorescent signal**. Fluorescent images were collected as indicated above, using identical parameters. Each data set contained an XYZ stack of GFP signals covering both sides of the calyx regions of MBs. The raw data were then processed using custom-made scripts in MATLAB. First, the raw data were loaded using the Bio-Formats library[67]. A 3D ROI covering only the unilateral calyx region was then manually selected in different XY planes along the Z axis. Subsequently, the fluorescence intensities of the voxels within the 3D ROI were averaged to represent the quantified fluorescent intensity of the whole unilateral calyx. As a result, each fly brain produced two independent calyx fluorescent intensities.

**Phototactic optogenetic stimulation**. Flies were reared at 18 °C and 60% humidity under darkness. A group of 40 flies were collected within 1 day after eclosion and transferred into a vial with regular food containing 200 mM all-trans retinal (Sigma R2500). The vials were wrapped in aluminium foil for protection from light, then kept at 25 °C and 60% humidity for 4 days. After transferral to the test dish, flies were allowed to recover for 5 min, and then stimulated with light. An array of white LEDs was used as the source of stimulation. Light stimulation was presented continually throughout the observation period. The light intensity was 28 mW/cm$^2$, measured using a spectrometer (CCS200/M, Thorlabs).

**Phototactic assay**. Phototactic choice behaviour was analyzed in a T maze[68]. Flies in groups of 40–50 were adapted in the dark for 1 min, then given 2 min to run toward the light before being trapped and counted. Phototaxis index = (number of flies in the tube close to the light − number of flies in the dark tube) /total number of flies.

**Odour preference assay**. The olfaction test was conducted in a T-maze[69]. Flies were starved in a tube, supplied with only water for 18–20 h before behavioral tests. Flies were first dark-adapted for 2 min in groups of 40–50, then given 10 min to choose between tubes containing apple cider vinegar (ACV) or water. Preference index = (number of flies in the tube containing ACV − number of flies in the control tube) / total number of flies.

**Movement ability assay**. The behavioural experiment was performed using a transparent 60-mm Petri dish filled with 1% agar to 5 mm below the lid. One

female fly was transferred into the dish. After 30 min of habituation, the movement of the flies in the dish was recorded for 4 h using a modified webcam at 1 s per frame. The velocity of the fly was calculated from the video using a MATLAB script.

**Social isolation treatment**. Third instar feeding larvae were housed in culture tubes either individually (singly reared) or with 20 siblings (group-reared). After eclosion, adult flies were transferred to fresh glass tubes and kept for 7 days before the start of the experiments.

**Surgery**. Surgery was performed to remove the primary olfactory organ (the third antennal segment) or sound detection organ (the arista on the third antennal segment) in *Drosophila*. Flies were anesthetized, and their corresponding peripheral organs were removed with a set of fine forceps. Flies were allowed to recover for 2 days before testing.

**Temperature-shift experiments**. Flies containing tubulin-Gal80$^{ts}$, UAS-TNT-E, and tissue-specific Gal4 transgenes were cultured at 18 °C. For the temperature-shift experiments, 1-day-old flies were transferred to and kept at 29 °C for 2 days. Before behavioural tests, flies were allowed to recover for 2 h at 25 °C to eliminate the interference of temperature stress on behaviour.

**Immunohistochemistry**. Fly brains were dissected in phosphate-buffered saline PBS, then fixed in 4% paraformaldehyde for 3 h on ice. Following four rinses in PBS containing 0.5% Triton X-100 (PBST), brains were blocked with 10% normal goat serum in PBST for 30 min at room temperature. Brains were then incubated with mouse anti-nc82 (1:100, DSHB) primary antibodies overnight at 4 °C. After washing four times, the brains were incubated with goat anti-mouse IgG con-jugated with TRITC (1:200, ZSGB-BIO) secondary antibodies overnight at 4 °C. Subsequently, the brains were mounted in Vectashield solution (Vector Labs Inc.). Fluorescent images were acquired using a Leica confocal microscope system[63].

**Statistics and reproducibility**. Statistical analyses were conducted using Prism 6 (GraphPad Software, Inc). Statistical analysis was conducted on data from five or more biologically independent experimental replicates. Analyzed numbers (n) from biologically independent samples are shown below each graph. Results are pre-sented as a box and whisker plot from multiple independent experiments. Dif-ferences between groups were analysed by Student's t-test (two-sided) or one-way ANOVA with Dunnett's test for post hoc analysis. $P > 0.05$ was considered non significant. $*P < 0.05$, $**P < 0.01$, and $***P < 0.001$ were considered significant.

**Reporting summary**. Further information on research design is available in the Nature Research Reporting Summary linked to this article.

## Data availability
The source data underlying Figs. 1c–e, 2a–e, 3b–h, 3j, 4b–d, 5b, 5d, 5g, 6a, 6f, 6g, 6i and Supplementary Figs. S1B–E, S1G–H, S2B–D, S3, S4B–C, S5, S6, S7, S8, S11, S12, S13D, S14, S15B, S15D, and S19C are provided as a Source Data file with this paper. Additional raw data that support the findings of this study are available upon request. Source data are provided with this paper.

## Code availability
MATLAB scripts used for analyses are available upon request.

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

## Acknowledgements

We thank the members of the Zhu lab for the helpful discussions. We are grateful to Dr. Yi Rao (Peking University) for critically reading our manuscript. We thank Meiyan Huang for technical assistance. We thank Dr. Aike Guo and Dr. Yan Li (Chinese Academy of Sciences), Dr. Yi Rao and Dr. Yulong Li (Peking University), Dr. Yi Zhong (Tsinghua University), Dr. Hongtao Qing (Hunan University), Dr. Yoshi Aso (Janelia Farm Research Campus), and Dr. Joshua Dubnau (Cold Spring Harbor Laboratory), as well as the *Drosophila* Genetic Resource Centre (DGRC), the Bloomington Stock Center (BSC), Core Facility of *Drosophila* Resource and Technology, and Tsinghua Fly Center for providing fly strains. This work was supported by NSFC grants (9163210042 and 31771173), the Beijing Advanced Discipline Fund, Key Research Program of Frontier Sciences of Chinese Academy of Sciences (CAS, QYZDY-SSW-SMC015), CAS Interdisciplinary Innovation Team, the Bill and Melinda Gates Foundation (OPP1119434) to Y.Z., and the Strategic Priority Research Program of CAS (Grant XDB02040002) to L.L.

## Author contributions

Y.S. and R.Q. conceived this project and designed all experiments., Y.S., R.Q., X.L. and F.K. performed most behavioural tests, Y.S., Q.R., Y.C. and S.G. analyzed the data, L.L. and Y.Z. supervised the project, Y.S. and Y.Z. wrote the paper.

## Competing interests

The authors declare no competing interests.
