## [Peer Review File · Nature Communications]

Reviewers' Comments:

Reviewer #1:

Remarks to the Author:

Comments on Sun et al.

Many animal species exhibit aggregation behavior, which may help sharing resources, finding mating partners, and averting predators. How this "sociality" arises is an interesting and important question in neuroscience. In the present manuscript, Zhu and colleagues examine neural mechanisms underlying social attraction in *Drosophila*. The authors developed an assay to test attraction to conspecifics, and show that a combined function of visual and olfactory systems is necessary for social attraction. A silencing screen identifies gamma mushroom body (MB) neurons to be essential for this behavior. A further examination indicates that a class of fan-shaped body (FB) neurons may transmit visual input, whereas the alpha/beta MB neurons may convey olfactory input. The authors suggest a circuit for social attraction, in which gamma MB neurons integrate FB and alpha/beta MB input. An additional experiment shows a role of serotonergic system in social attraction. Overall, I find the authors' results novel and interesting. However, there are several experimental and conceptual issues that I feel are necessary to be addressed before granting publication.

1) Specificity of drivers

The majority of drivers the authors use is enhancer-trap GAL4 lines, which lack precision in the specificity of neural population they label. For example, all five drivers that the authors used to label gamma neurons also exhibit expression in other MB neuron types as well as many neurons outside the MB (Table 2 and Figure S8). This lack of specificity likely contributes to results not easily interpretable, such as those presented in Figure S9A, in which silencing gamma neurons with 201Y driver, but not with NP1131 driver, leads to defect in phototaxis. A set of highly specific split-GAL4 drivers have been available especially for the MB neurons (Aso et al., 2014). I suggest that the authors repeat at least the essential set of experiments using split-GAL4 drivers. This may also allow authors to examine possible mechanisms of integration between vision and olfaction, because a subset of gamma neurons (i.e., gamma-d neurons) are known to receive visual input (Vogt et al., 2016).

2) Circuit model

The authors suggest that the gamma MB neurons integrate inputs from FB and alpha/beta MB neurons (Figure 4I). This is based on a GRASP experiment between FB and MB neurons (Figure 4H) and an experiment examining the localization of pre- and post-synaptic markers (Figure S11). These experiments are inadequate to support the circuit model. First, the GRASP experiment is done using the MB247 driver to express a GFP half. However, the MB247 driver labels all MB neurons (Riemensperger et al., 2005), and thus is not specific to gamma neurons. Second, the authors provide no evidence for the connection between alpha/beta and gamma neurons. To support this circuit model, it is necessary to perform more precise anatomical experiments, or (preferably) experiments that test functional connectivity. Without more convincing evidence, I would recommend the authors to de-emphasize this circuit model.

3) Role of serotonin

In the present format, the section describing a role of serotonin in social attraction appears disconnected from the main body of the manuscript. The authors describe expression of 5HT-1B receptor in gamma neurons - why not test requirement of this receptor in gamma neurons?

4) Emphasis on novelty

The authors emphasize that their manuscript establishes that *Drosophila* exhibits a high degree of (previously unappreciated) sociality. There has been well-known literature, however, that describes

aggregation behavior of *Drosophila* mediated by cis-vaccenyl acetate since Bartelt et al., 1985. I suggest that the authors acknowledge these previous studies.

5) Missing information and errors

The manuscript lacks critical information and contains numerous errors, especially in Materials and Methods and Figure Legends. For example, it is unclear how the preference index is calculated. The authors' description is "(time of flies spending on the side with tethered flies - time of flies spending on the side without flies) / total time". My assumption is that, if a single subject fly is used, then the preference index will be calculated as (number of video frames with this fly on the social side - on the other side) / total frames. But the authors mention that they only consider flies in the inner circle of the arena in order to account for thigmotaxis. Does this mean that a frame with the fly in the outer circle is considered not in the social side (i.e., include this frame in total)? What about when there are multiple subject flies? In addition, I notice several errors and missing information important for interpretation as well as reproducibility, some of which I list here.

- No description is provided for the result with the experiment using "fimo fly" (Figure S1F and G), in which the authors observe significant attraction to dummy flies made of clay and wings. What is the interpretation of the result?
- Fly strains used for 5-HT experiments (Figure 5) are not listed.
- What are the flies of genotypes "fruitless -/-" and "UAS-D2F-GFP"?
- Methods for denatonium conditioning (Figure S4C) are missing.
- Methods to quantify "social entrance index" (Figure S2C) are missing.
- In almost all figures, the authors say the data are presented as "mean +/- sem" but they use boxplot for most of the data presentation.
- Legend for Figure 5E is missing.
- Figure S1B and S1C look identical.

Reviewer #2:

Remarks to the Author:

The manuscript entitled "Social attraction in *Drosophila* is regulated by the mushroom body and serotonergic system" (NCOMMS-19-1723) seeks to establish a paradigm for studying sociality in *Drosophila* and demonstrate that this behavior is supported by the gamma-lobe of the mushroom bodies and regulated by serotonergic signaling. The study begins by characterizing several aspects of a behavioral assay in which flies will investigate other immobilized flies. Then using inactivation of a series of different sets of mushroom body lines, the study demonstrates a role for the gamma-lobe in the investigation of conspecifics. Finally, a role for the 5-HT1B receptor is proposed, as knocking down expression of this receptor interferes with investigation of conspecifics. This study describes a novel behavior/behavioral assay and the neural circuit underlying it. Despite this, I have several major concerns with this study that I would like to see addressed.

Major concerns.

-Definition of Sociality: The central and most significant claim in this study is that the authors have discovered a form of sociality in *Drosophila*, but no clear definition of "sociality" is provided or how the characteristics of this behavior qualify as "sociality". The authors need to either define sociality either as a trait with criteria that must be met or as existing along a spectrum. Whichever approach is used, there needs to be a discussion as to what constitutes "sociality", as there exist several definitions of sociality and the behavior in this study doesn't meet the criteria for some of these definitions. If the authors choose to provide an established set of criteria (rather than stating that sociality exists along a spectrum), they should make sure to point out how their behavioral assay demonstrates sociality as opposed to a stimulus evoked investigation of conspecifics. It may be easier to simply to

refer to this behavior as “social attraction” as they do in the title or “social investigation”.

-Mushroom body neurons: The authors frequently refer to “mushroom body neurons”. Do they mean Kenyon cells? If so, the authors should refer to them as Kenyon cells.

-CaLexA-NFAT experiments: The authors demonstrate that the conspecific recognition requires synaptic transmission by neurons in the gamma-lobe, but for the CaLexA-NFAT experiments they use a Gal4 line that they did not test in their behavioral experiments (BL46669), nor is it listed in the methods section for fly stocks. The more convincing data would be to at least show activation patterns in the gamma lobe. This data is being used to suggest that social interactions activate the mushroom bodies, but if a major claim is that social drive is processed in the gamma-lobe specifically, it would be important to demonstrate that the gamma-lobe is differentially activated relative to the other lobes. In theory, the authors should have this data, although perhaps both the surface alpha/beta as well as gamma kenyon cells are activated? Regardless, this would be good to include.

-use of UAS Tetanus-toxin: Why were UAS-TNT-IMP flies (the inactive version of this transgene) not included as control flies?

-Use of GRASP: The variant of GRASP used in this study has been demonstrated to produce false positives simply due to neurons coming into close proximity, but not actually having a functional synapse (MacPherson et al 2015). Furthermore, this version of GRASP does not allow for testing directionality of the synapse, so it cannot be claimed that the F5 neurons synapse upon Kenyon cells or vice versa. The transgenes used in figure S11 to highlight pre and postsynaptic sites are not sensitive enough reporters to be able to claim that neurons lack pre- or postsynaptic sites in a given region. Furthermore, the GRASP figures do not include a counterstain to delineate the processes of the cells themselves, so it is not possible to determine if this signal is legitimate. I would recommend staining for GFP with an antibody that only recognizes the GFP1-10 fragment to confirm that the GRASP signal localizes to the branches of the neurons of interest. Finally, the authors claim that gamma-lobe neurons integrate input from F5 neurons and alpha/beta Kenyon cells, but only test the F5 neurons. They should also demonstrate that the alpha/beta surface Kenyon cells provide synaptic input to gamma Kenyon cells.

-The authors should reference work in locusts demonstrating that aggregation behavior is induced by serotonin. This would seem to be extremely relevant, because locust phase shift involves animals changing from solitary to gregarious behavioral states in which they actively seek each others company.

-Use of general lines to inactivate serotonin cells and the 5-HT1B receptor: If the circuitry supporting this behavioral phenomenon is restricted to the gamma lobe of the mushroom body, why were manipulations of serotonin or the 5-HT1B receptor not directed to this brain region? DPM is the source of serotonin to the peduncles and lobes, but the authors used broadly expressed lines that lack expression in some serotonin cells and include some non-serotonergic neurons. Furthermore, if the gamma lobe is critical for this behavior why knock-down all expression of the 5-HT1B receptor rather than in only the Gal4 lines used in Figure 3?

-Fly strains used: The documentation for the fly lines used in this study is not sufficiently detailed. A table should be provided listing the genotype of flies used in each figure, as well as the source of those flies. For instance, there is no information as to the 5-HT receptor RNAi and Gal4 lines used in this study. The reader should be able to find sources and references for every fly line used in this study. This should include references that provide a complete demonstration of expression patterns of Gal4s and LexAs.

-Strength of claims: At some points in the discussion the authors make claims that are not supported by the data but can be rephrased to better reflect their results. On line 266 and 283 the authors claim that the mushroom bodies “promote” social behavior or are a “command center” for social behavior. This cannot be claimed, because the authors only showed that the mushroom bodies are required. To demonstrate that the mushroom body is a “command center”, you would have to activate it and induce social behavior to an object that normally does not elicit this behavior, such as flies of another species or small inanimate objects. This can be changed to say that these neurons are “required”. On line 270 the authors claim that social behavior is “preserved” between humans and flies, but this would suggest derivation from a common ancestor. Because this was not directly tested the authors should refer to this as a shared trait which avoids making any claim of homology vs. analogy. The same is true for the statement on line 295 about a conserved role between the hippocampus and mushroom bodies. Because organisms use multiple modalities to detect conspecifics with high fidelity, the integration of these modalities is likely an organizing principle common to all animals.

Minor concerns

- Line 58 should read “Regardless of the type of social”
- Line 91 should read “regardless of the social”
- Line 141 should read “to further examine the”
- The supplemental figure data are frequently presented completely out of order in relation to the text. For instance, in the results Fig. S4A is described first, then S4C and then finally S4B. Figure S2E is described in association with the data in figure S9, 4 pages after the remainder of figure S2 is described. Figure S5 is described before Figure S4. Please coordinate the order in which the figure panels and the text are presented.
- Line 149, please specify if male or female attractor flies were used for the fruitless experiments.
- Line 171 should read “other pairs of sensory modalities had no effect”
- Supplemental figure 8 uses a color scheme of red and green. Please adjust to magenta and green to accommodate red-green color blind readers.
- Line 269 should read “complicated behavioral traits”

{Comments from Reviewer #1}

Many animal species exhibit aggregation behavior, which may help sharing resources, finding mating partners, and averting predators. How this "sociality" arises is an interesting and important question in neuroscience. In the present manuscript, Zhu and colleagues examine neural mechanisms underlying social attraction in *Drosophila*. The authors developed an assay to test attraction to conspecifics, and show that a combined function of visual and olfactory systems is necessary for social attraction. A silencing screen identifies gamma mushroom body (MB) neurons to be essential for this behavior. A further examination indicates that a class of fan-shaped body (FB) neurons may transmit visual input, whereas the alpha/beta MB neurons may convey olfactory input. The authors suggest a circuit for social attraction, in which gamma MB neurons integrate FB and alpha/beta MB input. An additional experiment shows a role of serotonergic system in social attraction. Overall, I find the authors' results novel and interesting. However, there are several experimental and conceptual issues that I feel are necessary to be addressed before granting publication.

[Response to Reviewer 1]

We appreciate the reviewer's comments acknowledging the importance of this subject and the novelty of our study. The reviewer's comments helped us to substantially improve our manuscript. We have addressed the five main points raised by the reviewer as below.

[Comment #1] Specificity of drivers

The majority of drivers the authors use is enhancer-trap GAL4 lines, which lack precision in the specificity of neural population they label. For example, all five drivers that the authors used to label gamma neurons also exhibit expression in other MB neuron types as well as many neurons outside the MB (Table 2 and Figure S8). This lack of specificity likely contributes to results not easily interpretable, such as those presented in Figure S9A, in which silencing gamma neurons with 201Y driver, but not with NP1131 driver, leads to defect in phototaxis. A set of highly specific split-GAL4

drivers have been available especially for the MB neurons (Aso et al., 2014). I suggest that the authors repeat at least the essential set of experiments using split-GAL4 drivers. This may also allow authors to examine possible mechanisms of integration between vision and olfaction, because a subset of gamma neurons (i.e., gamma-d neurons) are known to receive visual input (Vogt et al., 2016).

[Response # 1]

► We agree that the enhancer-trap GAL4 lines lacked precision. In accordance with the reviewer's suggestion, we collected a set of split-GAL4 drivers (Aso et al., 2014) that are highly specific to the mushroom bodies (Fig. R1 below and Fig. 10 in the manuscript). We then suppressed the activities of these new Kenyon neurons with tetanus toxin. We found that social approach behaviour was only abolished following the expression of TNT in the KC_{γ} neurons of the mushroom body (labelled by either *MB607B-gal4*, *MB419B-gal4*, *MB009B-gal4*, or *MB131B-gal4*), whereas silencing other mushroom body neurons ($KC_{\alpha/\beta}$: *MB008B-gal4*, *MB477B-gal4*, *MB185B-gal4*, *MB594B-gal4* or *MB371B-gal4*; and $KC_{\alpha/\beta}$: *MB005B-gal4* or *MB463B-gal4*) did not affect social attraction (Fig. R2, and Fig. 3H in the manuscript). Taken together, these findings strongly indicated that γ lobe neurons are necessary for generating motivation for approaching conspecifics. Furthermore, these new data revealed that both γ_d neurons and γ_{main} neurons were required for promoting social attraction (Fig. R2, and Fig. 3H in the manuscript).

Fig. R2. KC_{γ} neurons mediated social motivation. The social approach levels were dramatically altered by UAS-TNT expression in *MB010B* ($\alpha/\beta+\alpha'/\beta'+\gamma$), *MB607B* (γ_d), *MB419B* (γ_d), *MB009B* ($\gamma_d+\gamma_{main}$), and *MB131B* ($\gamma_d+\gamma_{main}$) line-labelled neurons, but not other neurons (α'/β' : *MB005B-gal4* and *MB463B-gal4*; α/β : *MB008B-gal4*, *MB477B-gal4*, *MB185B-gal4*, *MB594B-gal4*, and *MB371B-gal4*). $n = 16-48$. The results are presented as a box and whisker plot; the whiskers indicate the minimum and maximum, the box includes the 25th–75th percentile, and the line in the box indicates the median of the data set. Statistical analysis: unpaired t-test. ns: $P > 0.05$, ***: $P < 0.001$.

[Comment #2] Circuit model

The authors suggest that the gamma MB neurons integrate inputs from FB and alpha/beta MB neurons (Figure 4I). This is based on a GRASP experiment between FB and MB neurons (Figure 4H) and an experiment examining the localization of pre- and post-synaptic markers (Figure S11). These experiments are inadequate to support the circuit model. First, the GRASP experiment is done using the MB247 driver to express a GFP half. However, the MB247 driver labels all MB neurons (Riemensperger et al., 2005), and thus is not specific to gamma neurons. Second, the authors provide no evidence for the connection between alpha/beta and gamma neurons. To support this circuit model, it is necessary to perform more precise anatomical experiments, or (preferably) experiments that test functional connectivity. Without more convincing evidence, I would recommend the authors to de-emphasize this circuit model.

[Response # 2]

We appreciate the reviewer's constructive suggestions. We conducted the suggested experiments, and the results were incorporated into the revised manuscript.

► We identified a specific line, *R72B08-Gal4*, and repeated the GRASP experiments with this line. The *R72B08* driver was expressed in multiple regions of the brain (Fig. R3A and R3B; Fig. S13A and S13B in the manuscript). However, in the mushroom body, this driver specifically labels KC_{γ} neurons (Fig. R3C and Fig. S13C in the

manuscript). Social approach behaviour was abolished following silencing of the *R72B08-Gal4*-labelled neurons with TNT, so *R72B08-Gal4* is functionally equivalent to the less specific *MB247-Gal4* (Fig. R3D and Fig. S13D in the manuscript).

We used an improved GRASP method to assess synaptic connectivity between KC γ neurons and F5 neurons. The targeted GFP reconstitution across synaptic partners (t-GRASP) method enhanced its specificity for synaptic contact sites using a targeting strategy for target GFP11 to pre-synaptic terminals and GFP1-10 to dendritic/post-synaptic regions (Shearin et al., 2018). We used *R84C10-lexA* to drive the expression of *LexAop-CD4::spGFP11* in F5 neurons (presynaptic, Fig. R4A and Fig. S17A-a in the manuscript) and the new *R72B08-Gal4* to drive the expression of *UAS-CD4::spGFP1-10* in KC γ neurons (postsynaptic, Fig. R4B and Fig. S17A-b in the manuscript). Only when the F5 neurons (*R84C10-LexA*) and KC γ neurons (*R72B08-Gal4*) separately expressed the complementary halves of GFP, intense labeling was observed near the calyx region (Fig. R4F-H and Fig. 5H b-d in the manuscript), demonstrating that F5 neurons synapse onto KC γ neurons in the brain.

Fig. R3. Blocking *R72B08-Gal4*-labelled neurons dramatically decreased the social approach level. (A and B) Expression patterns of *R72B08-GAL4* in the MB region were

visualized by *mCD8::GFP* (green). The neuropil was counterstained with the antibody against nc82 (red). Scale bar = 50 μ m. (C) Cross-sections of the expression patterns of *R72B08-GAL4* at peduncle, visualized by GFP. Magenta is the neuropil counterstaining with the antibody against nc82. Scale bar = 10 μ m. (D) Blocking *R72B08-GAL4* subsets of Kenyon cells impaired social motivation. $n = 32$. Results are presented as a box and whisker plot; the whiskers indicate the minimum and maximum, the box includes the 25th–75th percentile, and the line in the box indicates the median of the data set. Statistical analysis: unpaired t-test. ***: $P < 0.001$.

Fig. R4. Visualization of synaptic connections between $KC\gamma$ neurons (*R72B08-Gal4*) and F5 neurons (*R84C10-LexA*) in the calyx region. Expression of GFP driven by the *R84C10-LexA* driver (A) and the *R72B08-GAL4* driver (B) in the brain. (C) Cross-sections of the expression patterns of *R72B08-Gal4* at the peduncle. (D and E) No GFP signals in the negative controls for t-GRASP analysis. (F) t-GRASP signals indicate contacts between MB neurons (*R72B08-Gal4*) and F5 neurons (*R84C10-LexA*) in the calyx region (dashed box). (G and H) Magnified views of t-GRASP signals in dashed box regions of (F). Scale bars are 50 μ m, except in C (10 μ m) and in G–H (25 μ m). The neuropil was counterstained with an antibody against nc82 (magenta).

► To address the question regarding the connections between alpha/beta neurons and gamma neurons, we surveyed the downstream synaptic targets of alpha/beta neurons to other regions of the brain using *trans*-Tango, a method of anterograde transsynaptic tracing (Talay et al., 2017). In flies bearing the *NP3061-Gal4* driver (alpha/beta surface neurons) and the *trans*-Tango components, GFP-expressing $KC_{\alpha/\beta s}$ neurons innervated the α/β surface lobes as well as the α/β lobes and γ lobes, as indicated by distinct and robust mtdTomato labeling in corresponding MB regions (Fig. R5 and Fig. S18A in the manuscript). The later connectivity patterns indicated that $KC_{\alpha/\beta s}$ neurons transmit information to KC_{γ} neurons.

Furthermore, we analyzed the connectivity between $KC_{\alpha/\beta s}$ neurons and KC_{γ} neurons with the t-GRASP technique. We used *R44E04-lexA* to drive the expression of *LexAop-CD4::spGFP11* in $KC_{\alpha/\beta s}$ neurons (Fig. R6A and Fig. S18B-a in the manuscript) and *R72B08-Gal4* to drive the expression of *UAS-CD4::spGFP1-10* in KC_{γ} neurons (Fig. R6B and Fig. S18B-b in the manuscript). The *R44E04-lexA* driver specifically labelled $KC_{\alpha/\beta s}$ neurons in the mushroom body (Fig. R6C and Fig. S18B-c in the manuscript). Only when the $KC_{\alpha/\beta s}$ neurons (*R44E04-lexA*) and KC_{γ} neurons (*R72B08-Gal4*) separately expressed the complementary halves of GFP, intense labeling was observed near the calyx region (Fig. R6F-H and Fig. 5H f-h in the manuscript), demonstrating that $KC_{\alpha/\beta s}$ neurons synapse onto KC_{γ} neurons in the brain.

Fig. R5. Visualization of connections from $KC_{\alpha/\beta}$ neurons to KC_{γ} neurons by *trans*-Tango. In flies bearing the *trans*-Tango components, *NP3061-Gal4* drove the expression of ligand and myrGFP in α/β lobes of MB (A, green) and results in mtdTomato signals in postsynaptic α/β lobes and γ lobes of MB (B, magenta).

Fig. R6. Visualization of the connections between $KC_{\alpha\beta}$ neurons and KC_{γ} neurons in the calyx region. Expression patterns of *R44E04-LexA* ($KC_{\alpha\beta}$ neurons, A) and *R72B08-GAL4* (KC_{γ} neurons, B) in brains as visualized by GFP signals. (C) Cross-section showing the expression patterns of *R44E04* at the peduncle. (D and E) No GFP signals were found in the negative controls for t-GRASP analysis. (F) The t-GRASP signals indicate the contacts between KC_{γ} neurons (*R72B08-Gal4*) and $KC_{\alpha\beta}$ neurons (*R44E04-LexA*) in the calyx region (dashed box). (G and H) Magnified views of t-GRASP signals in the dashed box regions in (F). The neuropil was counterstained with an antibody against nc82 (magenta). Scale bars are 50 μm , except 10 μm in (C) and 25 μm in (G–H).

[Comment #3] Role of serotonin

In the present format, the section describing a role of serotonin in social attraction appears disconnected from the main body of the manuscript. The authors describe expression of 5HT-1B receptor in gamma neurons - why not test requirement of this receptor in gamma neurons?

[Response # 3]

► As suggested by the reviewer, we tested the requirement of 5HT1B receptors in gamma neurons using an RNAi knockdown approach. 5HT1B receptors were mainly expressed in KC_{γ} neurons of the mushroom body and the ellipsoid body neurons (Fig. R7A–C, Fig. 6E, and Fig. 6H a–c in the manuscript). When combined with a mushroom-body-specific *gal80* (*MB-Gal80*), Gal4-driven expression in the mushroom bodies was specifically eliminated (Fig. R7D-F and Fig. 6H d–f in the manuscript). Expressing the 5HT1B RNAi broadly in pan-neurons (by *elav-gal4*) or specifically in 5HT1B neurons (by *5HT1B-gal4*) decreased social approach behaviour (Fig. R7G and Fig. 6I in the manuscript), whereas preventing 5HT1B RNAi expression only in the KC neurons restored social approach behaviour in flies (Fig. R7G and Fig. 6I in the manuscript), indicating that normal social approach requires *5HT1B* receptors in the mushroom body.

Fig. R7. 5HT1B receptors in the mushroom bodies are required for motivation for social approach. Expression patterns were visualized by crossing *5HT1B-Gal4* to *UAS-CD8::GFP*. For each panel, a projection of a confocal stack of corresponding brain regions of a female fly is shown. (A–C) Expression patterns of the *5HT1B* receptor by *5HT1B-Gal4* in different brain regions: mushroom body (MB) region (A), ellipsoid body (EB) region (B), and whole-brain (C). (D–F) Expression patterns of 5HT1B receptors by *5HT1B-Gal4* in the presence of *MB-Gal80* in different brain regions: mushroom body region (D), ellipsoid body region (E), and whole-brain (F). Scale bar = 50 μ m. (G) RNAi Knockdown of 5HT1B expression in pan-neurons and 5HT1B

receptor neurons resulted in reduced social approach performance (gray bars). However, removing the expression of 5HT1B RNAi in the mushroom body by *MB-Gal80* prevented the social approach defect owing to the 5TH1B knockdown. n = 8. Results are presented as a box and whisker plot; the whiskers indicate the minimum and maximum, the box includes the 25th–75th percentile, and the line in the box indicates the median of the data set. Statistical analysis: unpaired t-test. ** P < 0.01, ***: P < 0.001.

[Comment #4] Emphasis on novelty

The authors emphasize that their manuscript establishes that *Drosophila* exhibits a high degree of (previously unappreciated) sociality. There has been well-known literature, however, that describes aggregation behavior of *Drosophila* mediated by cis-vaccenyl acetate since Bartelt et al., 1985. I suggest that the authors acknowledge these previous studies. (J Chem Ecol. 1985 Dec;11(12):1747-56. doi: 10.1007/BF01012124.)

[Response #4]

► We thank the reviewer for this helpful suggestion. In the revised manuscript, we included discussion of previous studies on aggregation (Aso et al., 2014) . Our new results indicated that cVA plays a role in social approach behaviour (Fig. R8 and Fig. S8 in the manuscript).

Fig. R8. cVA plays a partial role in social approach behaviour. Quantification of *Canton S* female flies attracted by virgin female attractor flies (eclosion for 4 hr: pink box; eclosion for 8 hr: red box), mated female attractor flies (light gray box), virgin male attractor flies (eclosion for 4 hr: green box; eclosion for 8 hr: blue box), or mated male attractor flies (dark gray box). n = 20-32. Results are presented as a box and whisker plot; the whiskers indicate the minimum and maximum, the box includes the 25th–75th percentile, and the line in the box indicates the median of the data set. Statistical analysis: unpaired t-test. *: P < 0.05, ***: P < 0.001.

[Comment #5] Missing information and errors

The manuscript lacks critical information and contains numerous errors, especially in Materials and Methods and Figure Legends. For example, it is unclear how the preference index is calculated. The authors' description is "(time of flies spending on the side with tethered flies - time of flies spending on the side without flies) / total time". My assumption is that, if a single subject fly is used, then the preference index will be calculated as (number of video frames with this fly on the social side - on the other side) / total frames. But the authors mention that they only consider flies in the inner circle of the arena in order to account for thigmotaxis. Does this mean that a frame with the fly in the outer circle is considered not in the social side (i.e., include this frame in total)? What about when there are multiple subject flies?

[Response #5]

► We thank the reviewer for pointing out these problems.

In the revised manuscript, we added a video to visually demonstrate how the preference index was calculated (Movie S1).

We used the following formula to calculate the performance index at each time point, then averaged the PI values over a designated period (4 hours).

$$PI = (N_f - N_0) / N_t$$

N_f : number of subject flies appearing on the side with tethered flies (inside of inner circle); N_0 : number of subject flies appearing on the side without tethered flies (inside

of inner circles); N_t : total number of subject flies (all flies in the arena). $N_t = 1$, when testing a single subject fly; $N_t = 5$, when testing multiple subject flies.

Regarding the reviewers' question on calculating PI with a single subject fly, a frame with the fly in the outer circle would have $N_f = 0$, $N_0 = 0$, and $N_t = 1$. For multiple subject flies, a frame with no fly on the social side inside of the inner circle would have $N_f = 0$ and $N_t = 5$, while the value of N_0 would depend on the number of flies on the empty side inside of the inner circle.

In addition, I notice several errors and missing information important for interpretation as well as reproducibility, some of which I list here.

- No description is provided for the result with the experiment using "fimo fly" (Figure S1F and G), in which the authors observe significant attraction to dummy flies made of clay and wings. What is the interpretation of the result?

▶ In accord with the reviewer's suggestion, we explained the "Fimo fly" method in more detail in the manuscript.

- Fly strains used for 5-HT experiments (Figure 5) are not listed.

▶ We thank the reviewer for this helpful suggestion. We listed all of the fly strains used in this study, including sources and identifiers, in an Excel file.

- What are the flies of genotypes "fruitless -/-" and "UAS-D2F-GFP"?

▶ We used *fru*^M null males (*fru*^{LexA}/*fru*⁴⁻⁴⁰) (Pan and Baker, 2014)
We used UAS-DenMark to replace the UAS-D2F line.

- Methods for denatonium conditioning (Figure S4C) are missing.

▶ An explanation of our method for quantifying "denatonium conditioning" were added to the revised "Materials and Methods" section.

- Methods to quantify "social entrance index" (Figure S2C) are missing.

▶ An explanation of our method for quantifying the “social entrance index” was added to the revised “Materials and Methods” section.

- In almost all figures, the authors say the data are presented as "mean +/- sem" but they use boxplot for most of the data presentation.

▶ The results are presented as a box and whisker plot; the whiskers indicate the minimum and maximum, the box includes the 25th–75th percentile, and the line in the box indicates the median of the data set.

- Legend for Figure 5E is missing.

▶ We added a legend for Figure 5E (Fig. 6E in the revised manuscript).

- Figure S1B and S1C look identical.

▶ We have corrected Figures S1B and S1C.

{Comments from Reviewer #2}

[Response to Reviewer 2]

We thank the reviewer for recognizing the value of our study. We also appreciate the reviewer's insightful comments and helpful suggestions.

[Comment #1] Definition of Sociality:

The central and most significant claim in this study is that the authors have discovered a form of sociality in *Drosophila*, but no clear definition of “sociality” is provided or how the characteristics of this behavior qualify as “sociality”. The authors need to either define sociality either as a trait with criteria that must be met or as existing along a spectrum. Whichever approach is used, there needs to be a discussion as to what constitutes “sociality”, as there exist several definitions of sociality and the behavior in this study doesn't not meet the criteria for some of these definitions. If the authors choose to provide an established set of criteria (rather than stating that sociality exists along a spectrum), they should make sure to point out how their behavioral assay demonstrates sociality as opposed to a stimulus evoked investigation of conspecifics. It may be easier to simply to refer to this behavior as “social attraction” as they do in the title or “social investigation”.

[Response #1]

► We thank the reviewer for their helpful suggestions.

We analyzed and quantified “social attraction” and “social investigation” behaviours using a social approach paradigm. However, our investigation was mainly focused on the underlying biological driving force(s) manifested as behaviour.

Compared with highly social insects, fruit flies lack advanced forms of sociality. Sociality in social ants and bees has been classified as eusociality (Nowak et al., 2010). In contrast, the social behaviour exhibited by fruit flies is traditionally considered to operate at the level of “pre-social” (Gadagkar, 1987), which essentially indicates anything “beyond the solitary.”

Previous reports by Simmon, Brug, Langan, and Naxh (Burg et al., 2013; Simon et al.,

2012), and previous studies in our lab have indicated that fruit flies spontaneously form aggregations without external cues. In the social approach paradigm, the strong tendency of freely-walking males (and females) to associate with immobilized males or females (Fig. 1C) suggests that this type of social attraction is more than courtship and mating. Furthermore, the long duration of association (1–4 hours, Fig. S1E) indicates that the behaviour is more than a simple social investigation.

In many social animals, sociality is typically referred to as a tendency to associate in social groups, and to form cooperative societies. The primitive social tendency in *Drosophila* enables us to investigate neural correlations without the complications of social structures. We focused on quantifying the group-forming tendency as an indicator of sociality, in accord with previous studies (Reiczigel et al., 2008), although other forms of group interactions and cooperation have also been reported to exist in *Drosophila* (Guo, 2017).

[Comment #2] Mushroom body neurons:

The authors frequently refer to “mushroom body neurons”. Do they mean Kenyon cells? If so, the authors should refer to them as Kenyon cells.

[Response #2]

► We thank the reviewer for this helpful suggestion. The mushroom body neurons were Kenyon cells; we have referred to them as Kenyon cells in the revised paper.

[Comment #3] use of UAS Tetanus-toxin: Why were UAS-TNT-IMP flies (the inactive version of this transgene) not included as control flies?

[Response #3]

► We thank the reviewer for raising this question. The TNT and IMP flies were generated using traditional P-element-mediated transgene methods. As the insertional sites were not controllable, the IMP is not a perfect genetic control for TNT.

We believed that it would be more informative to test a candidate Gal4 against other

Gal4s (as controls) by crossing them to TNT. Thus, in the revised manuscript, we compared the effect of one Gal4 (one group of neurons) vs. the other (other neurons), rather than comparing the effect of TNT vs. IMP.

As suggested by the reviewer, we investigated the role of DPM neurons (labelled by *C316-Gal4*) in social motivation. The results revealed that social approach behaviour was dramatically decreased when DPM neurons were silenced with TNT, whereas flies with the inactive form of TNT (*UAS-TNT^{imp}*) in DPM neurons exhibited normal social approach behaviour (Fig. R9C). This is an example to show that when *UAS-TNT^{imp}* is used, the behavioral phenotype is similar to that of wild type controls.

Fig. R9. Blocking DPM neurons substantially decreases the social approach level. (A and B) Expression patterns of *C316-GAL4* were visualized by mCD8::GFP (green). The neuropil was counterstained with the antibody against nc82 (magenta). Scale bar = 50 μ m. (C) Blocking *C316-GAL4* labelled serotonergic neurons impaired social motivation. $n = 40$. Results were presented as means \pm SEM. Statistical analysis: unpaired t-test. ***: $P < 0.001$.

[Comment #4] CaLexA-NFAT experiments:

The authors demonstrate that the conspecific recognition requires synaptic transmission by neurons in the gamma-lobe, but for the CaLexA-NFAT experiments they use a Gal4 line that they did not test in their behavioral experiments (BL46669), nor is it listed in

the methods section for fly stocks. The more convincing data would be to at least show activation patterns in the gamma lobe. This data is being used to suggest that social interactions activate the mushroom bodies, but if a major claim is that social drive is processed in the gamma-lobe specifically, it would be important to demonstrate that the gamma-lobe is differentially activated relative to the other lobes. In theory, the authors should have this data, although perhaps both the surface alpha/beta as well as gamma Kenyon cells are activated? Regardless, this would be good to include.

[Response #4] We thank the reviewer for their constructive comments.

► We obtained the *R72B08* (*BL46669*) driver at a very late stage of this project. Although *R72B08* drives expression in the mushroom body, antenna lobe, ellipsoid body, and other regions (Fig. R3A and B), this driver specifically labels KC_γ neurons in the mushroom body (Fig. R3C). Behavioural tests suggested that silencing *R72B08-Gal4*-labelled neurons abolished social approach behaviour (Fig. R3D), thereby making it an excellent choice for further investigations of circuitry and modulation. In the revised manuscript, we added *R72B08* to the list of fly stocks in the methods section.

As suggested by the reviewer, we conducted additional experiments to test whether gamma-lobe neurons are differentially activated, relative to the other lobes, by social interactions. We measured the specific activities of *R72B08*-labelled γ lobe neurons with CaLexA in flies with or without social experiences. Compared with socially isolated flies, group-reared flies exhibited higher fluorescence intensity in the γ lobe region (Fig. R10A and Fig. S14A in the manuscript) and in the calyx region of the mushroom body (Fig. 4A and B).

We also compared the activity of α/β surface neurons between isolated flies and group-reared flies using the CaLexA method. The group-reared flies exhibited higher, but not statistically significant, fluorescence intensities both in the α/β lobe region (Fig. R10B and Fig. S14B in the manuscript) and in the calyx region of MB (Fig. R10C and Fig. S14C in the manuscript) compared with socially isolated flies. Taken together, our

results strongly suggest that KC_γ neurons are a critical component of the neural circuits promoting social affiliation.

Fig. R10. (A) The fluorescent intensities via the CaLexA method were compared between the group-reared flies and singly-reared flies in the γ lobe region. Confocal images of KC_γ neurons in flies bearing *R72B08-GAL4*, *UAS-mLexA-VP16-NFAT*, *LexAop-CD2-GFP*, and *LexAop-CD8-GFP-2A-CD8-GFP* transgenes were analysed. $n = 10$. (B and C) The fluorescent intensities via the CaLexA method were compared between the group-reared flies and singly-reared flies in the α/β lobe region (B) and the calyx regions (C). Confocal images of $KC_{\alpha/\beta}$ neurons in flies bearing *NP3061-Gal4*, *UAS-mLexA-VP16-NFAT*, *LexAop-CD2-GFP*, and *LexAop-CD8-GFP-2A-CD8-GFP* transgenes were analysed. $n = 9-12$. Results are presented as a box and whisker plot; the whiskers indicate the minimum and maximum, the box includes the 25th–75th percentile, and the line in the box indicates the median of the data set. Statistical analysis: unpaired t-test.

[Comment #5] The authors should reference work in locusts demonstrating that

aggregation behavior is induced by serotonin. This would seem to be extremely relevant, because locust phase shift involves animals changing from solitary to gregarious behavioral states in which they actively seek each others company

[Response #5]

► We thank the reviewer for this helpful suggestion. In the revised manuscript, we reviewed and discussed previous studies of serotonin on aggregation in locusts.

[Comment #6] Use of general lines to inactivate serotonin cells and the 5-HT1B receptor:

If the circuitry supporting this behavioral phenomenon is restricted to the gamma lobe of the mushroom body, why were manipulations of serotonin or the 5-HT1B receptor not directed to this brain region? DPM is the source of serotonin to the peduncles and lobes, but the authors used broadly expressed lines that lack expression in some serotonin cells and include some non-serotonergic neurons. Furthermore, if the gamma lobe is critical for this behavior why knock-down all expression of the 5-HT1B receptor rather than in only the Gal4 lines used in Figure 3?

[Response #6]

► We thank the reviewer for these helpful suggestions. We conducted additional experiments in accord with the reviewer's comments.

First, as suggested, we tested more specific DPM neurons, which were labelled by *C316-Gal4*, in accord with a previous study (Lee et al., 2011). *C316* exhibited strong expression in the lobes and peduncles of the mushroom body (Fig. R9A-B and Fig. S19A-B in the manuscript). Social approach behaviour was substantially decreased when DPM neurons were silenced by TNT (Fig. R9C and Fig. S19C in the manuscript). This result indicated that these serotonergic neurons innervating the mushroom body are responsible for normal social approach behaviour.

Second, we analyzed the effects of knocking-down the expression of 5-HT1B only in the gamma lobe. The 5-HT1B receptors were enriched in KC_{γ} neurons in the mushroom

body, and ellipsoid body neurons (Fig. R7A–C, Fig. 6E, and Fig. 6H a–c in the manuscript). We took advantage of *MB-Gal80*, which specifically blocks Gal4-driven gene expression only in MB regions (Fig. R7D–F and Fig. 6H d–f in the manuscript). We found that restricting the expression of 5HT1B RNAi in KC neurons restored social approach behaviour in *elav>5HT1B^{RNAi}* flies and *5HT1B>5HT1B^{RNAi}* flies (Fig. R7G and Fig. 6I in the manuscript), indicating that social approach behaviour specifically requires the 5HT1B receptor in the mushroom body, but not in the other tested regions.

[Comment #7] Fly strains used:

The documentation for the fly lines used in this study is not sufficiently detailed. A table should be provided listing the genotype of flies used in each figure, as well as the source of those flies. For instance, there is no information as to the 5-HT receptor RNAi and Gal4 lines used in this study. The reader should be able to find sources and references for every fly line used in this study. This should include references that provide a complete demonstration of expression patterns of Gal4s and LexAs.

[Response #7]

► We thank the reviewer for this helpful suggestion. We listed all of the fly strains used in this study, including sources and identifiers, in an Excel file.

[Comment #8] Strength of claims: At some points in the discussion the authors make claims that are not supported by the data but can be rephrased to better reflect their results. On line 266 and 283 the authors claim that the mushroom bodies “promote” social behavior or are a “command center” for social behavior. This cannot be claimed, because the authors only showed that the mushroom bodies are required. To demonstrate that the mushroom body is a “command center”, you would have to activate it and induce social behavior to an object that normally does not elicit this behavior, such as flies of another species or small inanimate objects. This can be changed to say that these neurons are “required”. On line 270 the authors claim that

social behavior is “preserved” between humans and flies, but this would suggest derivation from a common ancestor. Because this was not directly tested the authors should refer to this as a shared trait which avoids making any claim of homology vs. analogy. The same is true for the statement on line 295 about a conserved role between the hippocampus and mushroom bodies. Because organisms use multiple modalities to detect conspecifics with high fidelity, the integration of these modalities is likely an organizing principle common to all animals.

[Response #8]

► We thank the reviewer for this critical question and helpful advice.

First, to address the reviewer’s concern regarding the “command center” (lines 266 and 283 in the previous version of the manuscript), we artificially activated KC_γ neurons labelled with *NP1131-Gal4* or *R72B04-Gal4* using an optogenetic method. Activating KC_γ neurons evoked greater social motivation between individuals of the same species (conspecific, *Canton-S*, Fig. R12A and Fig. 4C in the manuscript). Interestingly, activating KC_γ neurons also dramatically increased the level of social approach toward different species (*D. repleta*, Fig. R12B and Fig. 4D in the manuscript). Taken together, these results strongly suggested that these KC_γ neurons are a critical component of the neural circuits promoting social affiliation.

Second, regarding our discussion of the relationship between social approach behaviour in flies and that in other species (lines 270 and 295 in the previous version), we have revised the text carefully to ensure the accuracy of our claims.

Fig. R12. Activating KC_{γ} neurons evokes social motivation in *Drosophila*. (A) In the test of social interaction between the same species (*Canton S*), optogenetic activation of *NP1131-GAL4*- or *R72B04-Gal4*-labelled Kenyon cells resulted in higher levels of social approach. $n = 24$. (B) In the test of social interaction between different species (*Canton S* vs. *D. repleta*), activating *NP1131-GAL4*- or *R72B04-Gal4*-labelled Kenyon cells also generated higher levels of social approach. $n = 16-24$. The results are presented as a box and whisker plot; the whiskers indicate the minimum and maximum, the box includes the 25th–75th percentile, and the line in the box indicates the median of the data set. Statistical analysis: One-way ANOVA followed by Dunnett’s test comparing all boxes vs. control box. ns: $P > 0.05$, *: $P < 0.05$, **: $P < 0.01$, ***: $P < 0.001$.

[Comment #9] Minor concerns

-Line 58 should read “Regardless of the type of social”

► We thank the reviewer for this helpful suggestion. We changed “Deficits in social approach” to “Deficits in social” in the revised manuscript.

-Line 91 should read “regardless of the social”

► We thank the reviewer for this helpful suggestion. We changed “regardless the

gender of attractors” to “regardless of the social” in the revised manuscript.

-Line 141 should read “to further examine the”

► We thank the reviewer for this helpful suggestion. We changed “to further exam the” to “to further examine the” in the revised manuscript.

-The supplemental figure data are frequently presented completely out of order in relation to the text. For instance, in the results Fig. S4A is described first, then S4C and then finally S4B. Figure S2E is described in association with the data in figure S9, 4 pages after the remainder of figure S2 is described. Figure S5 is described before Figure S4. Please coordinate the order in which the figure panels and the text are presented.

► We thank the reviewer for this helpful suggestion. We carefully coordinated the order in which the figure panels and text are presented.

-Line 149, please specify if male or female attractor flies were used for the fruitless experiments.

► We thank the reviewer for this helpful suggestion. Male flies were used for the fruitless experiments. We added this information in both the revised “Materials and Methods” section and the revised “Figure legends” section.

-Line 171 should read “other pairs of sensory modalities had no effect”

► We thank the reviewer for this helpful suggestion. We changed “other pairs of sensory pairs had no effect” to “other pairs of sensory modalities did not affect”.

-Supplemental figure 8 uses a color scheme of red and green. Please adjust to magenta and green to accommodate red-green color blind readers.

► We thank the reviewer for this helpful suggestion. We adjusted red and green to magenta and green.

-Line 269 should read “complicated behavioral traits”

► We thank the reviewer for this helpful suggestion. We changed “complicated behavio

ural traits” to “complex behavioural traits”.

Reference:

Aso, Y., Sitaraman, D., Ichinose, T., Kaun, K.R., Vogt, K., Belliart-Guerin, G., Placais, P.Y., Robie, A.A., Yamagata, N., Schnaitmann, C., *et al.* (2014). Mushroom body output neurons encode valence and guide memory-based action selection in *Drosophila*. *eLife* 3.

Burg, E.D., Langan, S.T., and Nash, H.A. (2013). *Drosophila* social clustering is disrupted by anesthetics and in narrow abdomen ion channel mutants. *Genes Brain and Behavior* 12, 338-347.

Gadagkar, R. (1987). What are social insects? *Indian Chapter Newsletter* 1 (20).

Guo, A., Zhefeng Gong, Hao Li, Yan Li, Li Liu, Qingqing Liu, Huimin Lu, Yufeng Pan, Qingzhong Ren, Zhihua Wu, Ke Zhang & Yan Zhu (2017). "1.27 - Vision, Memory, and Cognition in *Drosophila*." In *Learning and Memory: A Comprehensive Reference* (Second Edition), edited by John H. Byrne. 483-503.

Lee, P.T., Lin, H.W., Chang, Y.H., Fu, T.F., Dubnau, J., Hirsh, J., Lee, T., and Chiang, A.S. (2011). Serotonin-mushroom body circuit modulating the formation of anesthesia-resistant memory in *Drosophila*. *Proceedings of the National Academy of Sciences of the United States of America* 108, 13794-13799.

Nowak, M.A., Tarnita, C.E., and Wilson, E.O. (2010). The evolution of eusociality. *Nature* 466, 1057-1062.

Pan, Y.F., and Baker, B.S. (2014). Genetic Identification and Separation of Innate and Experience-Dependent Courtship Behaviors in *Drosophila*. *Cell* 156, 236-248.

Reiczigel, J., Lang, Z., Rozsa, L., and Tothmeresz, B. (2008). Measures of sociality: two different views of group size. *Anim Behav* 75, 715-721.

Shearin, H.K., Quinn, C.D., Mackin, R.D., Macdonald, I.S., and Stowers, R.S. (2018). t-

GRASP, a targeted GRASP for assessing neuronal connectivity. *Journal of Neuroscience Methods* *306*, 94-102.

Simon, A.F., Chou, M.T., Salazar, E.D., Nicholson, T., Saini, N., Metchev, S., and Krantz, D.E. (2012). A simple assay to study social behavior in *Drosophila*: measurement of social space within a group. *Genes Brain and Behavior* *11*, 243-252.

Talay, M., Richman, E.B., Snell, N.J., Hartmann, G.G., Fisher, J.D., Sorkac, A., Santoyo, J.F., Chou-Freed, C., Nair, N., Johnson, M., *et al.* (2017). Transsynaptic Mapping of Second-Order Taste Neurons in Flies by trans-Tango. *Neuron* *96*, 783-+.

Reviewers' Comments:

Reviewer #1:

Remarks to the Author:

Sun et al.

I find the manuscript much stronger and more convincing, thanks to the authors' efforts in addressing all of the concerns I had previously. In fact, I have read it with excitement.

One serious concern I have is the logic for the authors' assessment of the role of cVA in their behavioral assay (note that this section is added for the revision). The authors use virgin females as tethered target flies and observed a diminished social attraction as compared to mated females as target (lines 199-204, Figure S8). Their logic is that because virgin females lack cVA, the observed decrease in social approach is attributable to the lack of cVA. In order to show that this effect is indeed mediated by cVA, however, the authors would need to perfume cVA on virgin females and see whether this would rescue social approach. Or even better, the authors should examine mutant flies that lack the cVA receptor (i.e., Or67d mutants) as subject flies to examine the role of cVA in social approach. While these would be interesting experiments for the future, for now, I suggest that the authors revise this section in such a manner that accommodates the caveat of their experiments in linking cVA and social approach.

I have two suggestions for discussion. The additional experiments performed using split-GAL4 drivers have not only substantiated the authors' conclusions, but also provided an intriguing observation that the activity of gamma-d Kenyon cells (KCs), which receive visual input (Vogt et al., 2016), is required for social attraction (Figure 3H, MB607B and MB419B). I feel that the authors should discuss implication of this interesting finding.

The input neuropil of the mushroom body, calyx, is divided into main and accessory calyces and the gamma-d KCs receive visual input in the ventral accessory calyx (Yagi et al., 2016 and Vogt et al., 2016). The authors' t-GRASP experiments suggest the presence of synapses between gamma KCs and F5 neurons (Figure 5H) "near the calyx region" (lines 295). Are these synapses in the ventral accessory calyx?

Finally, I list four points that I feel are important to further improve the manuscript.

1. The authors should provide statistics comparing attraction toward real flies vs Fimo flies to substantiate their claim that attraction to real flies is higher (line 114).
2. The authors might want to include the finding from Mercier et al. (2018, Current Biology) in their description for the role of cVA as aggregation pheromone (line 197). Mercier provides a convincing evidence that cVA acts as attractant.
3. I find the sentences in lines 249-251 misleading, claiming that the alpha/beta KCs are more active in group-rearing than in social isolation, then revealing that this effect is not statistically significant. I suggest that the authors either indicate statistical non-significance upfront or eliminate these sentences altogether.
4. I suggest that the authors mention silencing F5 neurons in intact flies does not cause social attraction defect (Fig. 5G) before lines 268-271, in which they describe the effect of silencing F5 neurons in anosmic flies.

Reviewer #2:

Remarks to the Author:

I applaud the authors for systematically addressing each of my concerns with extensive and thorough follow up experiments. They have described an intriguing behavioral paradigm and provided convincing mechanistic data to implicate the role of the gamma lobe as a multi-sensory integrator underlying social interactions. I have no further comments and congratulate them on producing an excellent study.

REVIEWERS' COMMENTS:

Reviewer #1 (Remarks to the Author):

Sun et al.

I find the manuscript much stronger and more convincing, thanks to the authors' efforts in addressing all of the concerns I had previously. In fact, I have read it with excitement.

We appreciate the reviewer's positive comments.

One serious concern I have is the logic for the authors' assessment of the role of cVA in their behavioral assay (note that this section is added for the revision). The authors use virgin females as tethered target flies and observed a diminished social attraction as compared to mated females as target (lines 199-204, Figure S8). Their logic is that because virgin females lack cVA, the observed decrease in social approach is attributable to the lack of cVA. In order to show that this effect is indeed mediated by cVA, however, the authors would need to perfume cVA on virgin females and see whether this would rescue social approach. Or even better, the authors should examine mutant flies that lack the cVA receptor (i.e., Or67d mutants) as subject flies to examine the role of cVA in social approach. While these would be interesting experiments for the future, for now, I suggest that the authors revise this section in such a manner that accommodates the caveat of their experiments in linking cVA and social approach.

► We appreciate the reviewer's constructive suggestions.

As suggested, to further confirm that cVA was involved in social approach behaviour, we analysed the behaviour of flies without functional cVA-sensing neurons. There are two kinds of cVA receptor neurons in *Drosophila*: Or67d neurons (labelled by *Or67d*-

Gal4) and Or65a neurons (labelled by *Or65a-Gal4*). Flies with the cVA receptor neurons silenced by TNT were tested in the dark for social affiliation ability. The social approach response was strongly suppressed when Or67d receptor neurons were silenced, but not when Or65a receptor neurons were silenced (Fig. R1), suggesting that cVA signalling mediated by Or67d receptor neurons is important in the social approach paradigm. This result is in accord with the finding by Mercier et al. 2018 (as suggested by the reviewer) that Or67d neurons serve as receptors for tracking male-deposited landmarks. These new data strengthen our previous conclusion that cVA plays a role in social approach behaviour.

In the revised manuscript, we modified the corresponding text in the cVA section to introduce the findings by Mercier et al. 2018 and incorporate our new results (also in Fig. S8B).

Fig. R1. cVA plays a partial role in social approach behaviour. Blocking *Or76d-GAL4* labelled olfactory receptor neurons impaired social motivation. Results are presented as a box and whisker plot; the whiskers indicate the minimum and maximum, the box includes the 25th–75th percentile, and the line in the box indicates the median of the data set. Analyzed numbers (n) from biologically independent samples are showed below each graph. Statistical analysis: unpaired t-test. ns: $P > 0.05$, **: $P < 0.01$, ***: $P < 0.001$.

< 0.001.

I have two suggestions for discussion.

The additional experiments performed using split-GAL4 drivers have not only substantiated the authors' conclusions, but also provided an intriguing observation that the activity of gamma-d Kenyon cells (KCs), which receive visual input (Vogt et al., 2016), is required for social attraction (Figure 3H, MB607B and MB419B). I feel that the authors should discuss implication of this interesting finding. The input neuropil of the mushroom body, calyx, is divided into main and accessory calyces and the gamma-d KCs receive visual input in the ventral accessory calyx (Yagi et al., 2016 and Vogt et al., 2016). The authors' t-GRASP experiments suggest the presence of synapses between gamma KCs and F5 neurons (Figure 5H) "near the calyx region" (lines 295). Are these synapses in the ventral accessory calyx?

► We thank the reviewer for this helpful suggestion. Indeed, our results in Fig. 3H suggested that gamma-d Kenyon cells ($KC_{\gamma d}$ neurons, labelled by *R72B08-Gal4*) mediate visual information during social approach.

As the reviewer pointed out, the calyx is divided into main and accessory calyces, which receive different inputs. It has previously been shown that olfactory inputs project to the main calyx while visual stimuli project to the accessory calyx (Vogt et al., 2016, and Yagi et al., 2016). In accord with these previous studies, our t-GRASP experiments here indicated that social cues of distinct sensory modalities are presented to different KC subsets in subdomains of the calyx. The t-GRASP signals, which are indicative of synaptic sites between the gamma KCs and F5 neurons, were detected in the accessory calyx (Fig. 5Hb-d and Fig. S17Bf-h).

We modified the text in the revised manuscript to reflect this observation (lines 337-344).

Finally, I list four points that I feel are important to further improve the manuscript.

1. The authors should provide statistics comparing attraction toward real flies vs Fimo flies to substantiate their claim that attraction to real flies is higher (line 114).

► We thank the reviewer for this suggestion. The statistical comparison between real flies vs fimo flies has been added to the revised manuscript in Fig. S1G.

2. The authors might want to include the finding from Mercier et al. (2018, Current Biology) in their description for the role of cVA as aggregation pheromone (line 197). Mercier provides a convincing evidence that cVA acts as attractant.

► We thank the reviewer for this helpful suggestion. The finding by Mercier et al. 2018 serves as an excellent introduction to test cVA signalling in the social approach paradigm. Our new results on *Or67d* receptor neurons (Fig. R1 and Fig. S8B in revised manuscript) are in accord with the role of *Or67d* ORNs suggested by Mercier et al. 2018.

We modified the cVA section to incorporate the findings by Mercier et al. 2018 in the revised manuscript.

3. I find the sentences in lines 249-251 misleading, claiming that the alpha/beta KCs are more active in group-rearing than in social isolation, then revealing that this effect is not statistically significant. I suggest that the authors either indicate statistical non-significance upfront or eliminate these sentences altogether.

► We thank the reviewer for pointing out these problems. As suggested, we have indicated statistical non-significance upfront in the revised manuscript (lines 267-270):“We also compared fluorescent signals in other regions between group-reared flies and socially isolated flies. The differences in either the α/β lobe region (Fig. S14B) or the calyx region (Fig. S14C) were not statistically significant.”

4. I suggest that the authors mention silencing F5 neurons in intact flies does not cause social attraction defect (Fig. 5G) before lines 268-271, in which they describe the effect of silencing F5 neurons in anosmic flies.

► The result that silencing F5 neurons in intact flies did not cause social attraction defects was already mentioned in Fig. 4D and lines 210–213, before Fig. 5G, and lines 268–271.

Reviewer #2 (Remarks to the Author):

I applaud the authors for systematically addressing each of my concerns with extensive and thorough follow up experiments. They have described an intriguing behavioral paradigm and provided convincing mechanistic data to implicate the role of the gamma lobe as a multi-sensory integrator underlying social interactions. I have no further comments and congratulate them on producing an excellent study.

We thank the reviewer for appreciating our work and recognizing the value of our study.

References:

Mercier D, Tsuchimoto Y, Ohta K, Kazama H. (2018) Olfactory Landmark-Based Communication in Interacting *Drosophila*. *Curr Biol*. 2018 Aug 20;28(16):2624-2631.e5.

Yagi R, Mabuchi Y, Mizunami M, Tanaka NK. (2016) Convergence of multimodal sensory pathways to the mushroom body calyx in *Drosophila melanogaster*. *Sci Rep*. 2016 Jul 11;6:29481.

Vogt K, Aso Y, Hige T, Knapek S, Ichinose T, Friedrich AB, Turner GC, Rubin GM, Tanimoto H. (2016) Direct neural pathways convey distinct visual information to *Drosophila* mushroom bodies. *Elife*. 2016 Apr 15;5:e14009.